# Scaffolds with optimized quaternary symmetry for de novo cryoEM structure determination of small RNAs

**Christopher P. Jones** ✉ **& Adrian R. Ferré-D'Amaré** ✉

Structured RNAs play many roles in cells and emerging biotechnology. While large RNAs and ribonucleoprotein complexes often benefit from high-resolution structural analysis through cryogenic-sample electron microscopy (cryoEM), single-domain RNAs, particularly those smaller than ~100 nt (33 kDa), have proven challenging. Here we address this methodological gap by engineering two- and fourfold symmetric scaffolds that enable de novo structure solution of covalently attached RNA guests to beyond 3 Å overall resolution for the best resolved guests. We apply C2 and D2-symmetric scaffolds to post-transcriptionally unmodified tRNA$^{Asp}$, the fluorogenic aptamer Mango-III, and previously uncharacterized quinine- and 8-oxoguanine-binding aptamers. Experimental Coulomb potential maps with quality sufficient for small-molecule ligand, cation and water molecule placement reveal the molecular basis for specificity and suggest routes for structure-guided RNA engineering. Optimized scaffolds with intrinsic quaternary structure are a new general tool to interrogate the atomistic architecture of natural and designed compact RNA folds by single-particle cryoEM.

In recent years, single-particle cryogenic-sample electron microscopy (cryoEM) has enabled structure determination of specimens not tractable by other methods, including multistate RNA–protein complexes such as the spliceosome[1–3]. As the frontiers of technical feasibility get extended to smaller and more dynamic samples[4–7], a substantial technical gap remains in obtaining high-resolution structures of compact single-domain RNAs, in particular those that recognize small-molecule ligands. Preparation of well-ordered RNA crystals can be challenging, often necessitating diverse, labor-intensive screening and post-crystallization treatment strategies[8–10]. Consequently, successful application of cryoEM to determine new RNA structures without requiring crystallization is a highly sought objective.

Here, we demonstrate the utility of a compact RNA domain that adopts symmetric quaternary structures as a scaffold or host molecule to enable structure determination for RNAs otherwise too small for cryoEM. Our approach synergistically combines the strengths of two previously described heuristics for RNA cryoEM. As summarized in Supplementary Table 1, published scaffold approaches[5,11–13] use large RNAs of known structure as hosts into which guest RNAs of interest are attached covalently. By employing Group I or Group II self-splicing introns, structures at various resolutions of host RNAs have been determined[11,12]; however, the host–guest complexes become very large because of the host RNA sizes (250–400 nt). The presence of symmetry in cryoEM samples benefits both data collection and analysis by increasing particle size and copy number and by improving particle alignment. By engineering intermolecular contacts into RNAs of known structure[14], twofold symmetric particles yielded superior cryoEM structures compared to the parental monomers. Those experiments[14,15] demonstrated the utility of symmetry in RNA cryoEM but are limited by needing to know the three-dimensional (3D) structure of the target RNA before engineering symmetric intermolecular interactions. Our approach employs a compact (~100 nt) host RNA that intrinsically oligomerizes. Thus, our compact scaffold itself provides quaternary symmetry without having to engineer the guest whose structure, typically unknown a priori, is of interest.

Laboratory of Nucleic Acids, National Heart, Lung and Blood Institute, Bethesda, MD, USA. ✉e-mail: christopher.jones2@nih.gov; adrian.ferre@nih.gov

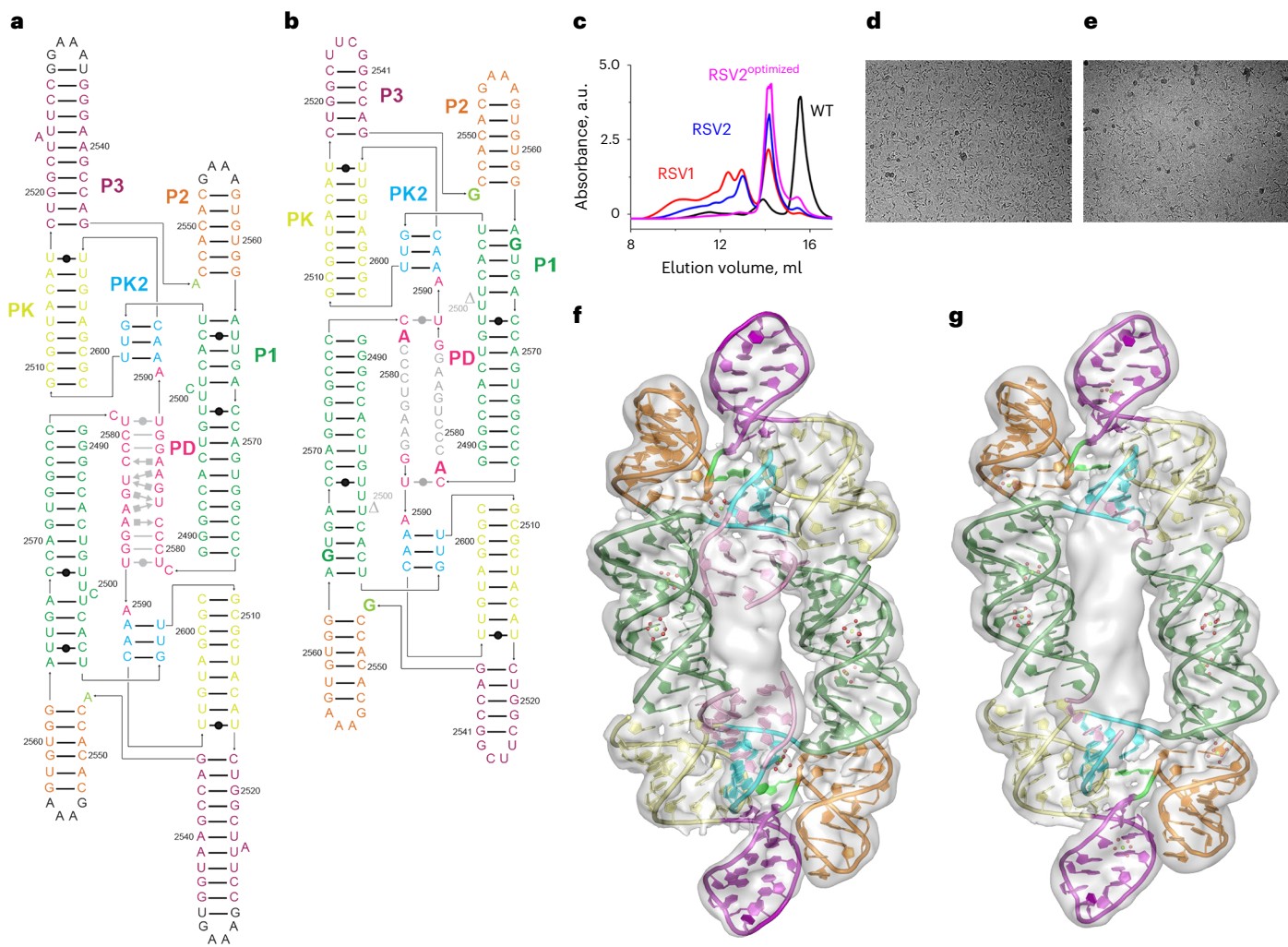

**Fig. 1 | Engineering of twofold optimized host RNAs. a**, Secondary structure of the dimeric RSV pseudoknot from the previously published cryoEM structure[19]. Black and gray Leontis–Westhof symbols[39] denote intra- and intermolecular base pairing, respectively, and thin lines indicate connectivity. Residue numbers correspond to RSV Prague C strain genome numbering. Residues previously changed for crystallization are black. **b**, Secondary structure of optimized dimeric RNA RSV2 ('2OP'). Mutations are indicated by bold letters, and gray letters indicate unmodeled residues. 'D' indicates the deletion of C2500. **c**, Size-exclusion chromatogram (absorbance at 254 nm) of various RNAs:

parental RSV FSE ('WT', black), RSV1 (red), RSV2 (blue) and RSV2 with optimized folding (magenta). The void volume is 8 ml. **d,e**, Example micrographs of RSV1 (**d**) and RSV2 (**e**) particles. **f**, Map and structure of RSV1, contoured at 10 σ in Pymol and overlaid with the structure of RSV1 shown as a cartoon. Color scheme corresponds to **a** and **b**. A2546 is colored light green. Pink and green spheres denote water molecules and magnesium ions, respectively. **g**, Map and structure of 2OP, colored as in **b**, contoured at 10 σ in Pymol, and overlaid with the structure of RSV2 shown as a cartoon.

Although uncommon in the structural database[16], several naturally occurring oligomeric RNAs with symmetric quaternary structures have been discovered in viruses, in which they dimerize retroviral genomes[6,17] or package viral RNA into phage[18]. Recently, we determined[19] the structure of the Rous sarcoma virus (RSV) frameshifting stimulatory element (FSE) by both X-ray crystallography and cryoEM. Functionally a monomer, crystals of this RNA contained a dimer of two nonidentical conformers in the crystallographic asymmetric unit; in contrast, the cryoEM specimen contained a fraction of symmetric dimers. The latter serendipitously facilitated cryoEM structure determination[19]. Now, we have engineered the RSV FSE to create host RNAs that are majority dimeric in solution and on the cryoEM grid. Notably, we observed these C2-symmetric RNAs occasionally form D2-symmetric tetramers (dimers of dimers), which further enhanced cryoEM analysis. We applied our optimized host scaffolds to various covalently attached guest RNAs of known structure[20,21] that include unmodified *Escherichia coli* tRNA[Asp] and the fluorogenic aptamer iMango-III A10U. Furthermore, we used this method to determine the

structures of the recently discovered synthetic aptamer for quinine[22] and natural aptamer for 8-oxoguanine (8-oxoG)[23]. For both of these, chains were traced de novo, and map quality was sufficiently high to define interactions within ligand-binding pockets that determine their respective specificities. In addition to providing a powerful new tool for RNA structural biology, our work outlines principles to guide future development of nucleic acid guest molecules with optimized quaternary structure for cryoEM structure determination of compact DNA and RNA domains and their bound small-molecule ligands.

## Results

### Design of host RNA scaffolds with optimized dimer yield

The starting point for design was the RSV FSE, of which ~15% forms dimers both in solution and on the cryoEM grid[19]. Each protomer is composed of five paired elements that include two pseudoknots (P1, P2, P3, PK1 and PK2) (Fig. 1a). Two such chains associate through a dimerization helix (PD) (Fig. 1a). To enhance quaternary structure formation, the parental RNA was altered by truncating P3 (Fig. 1b and

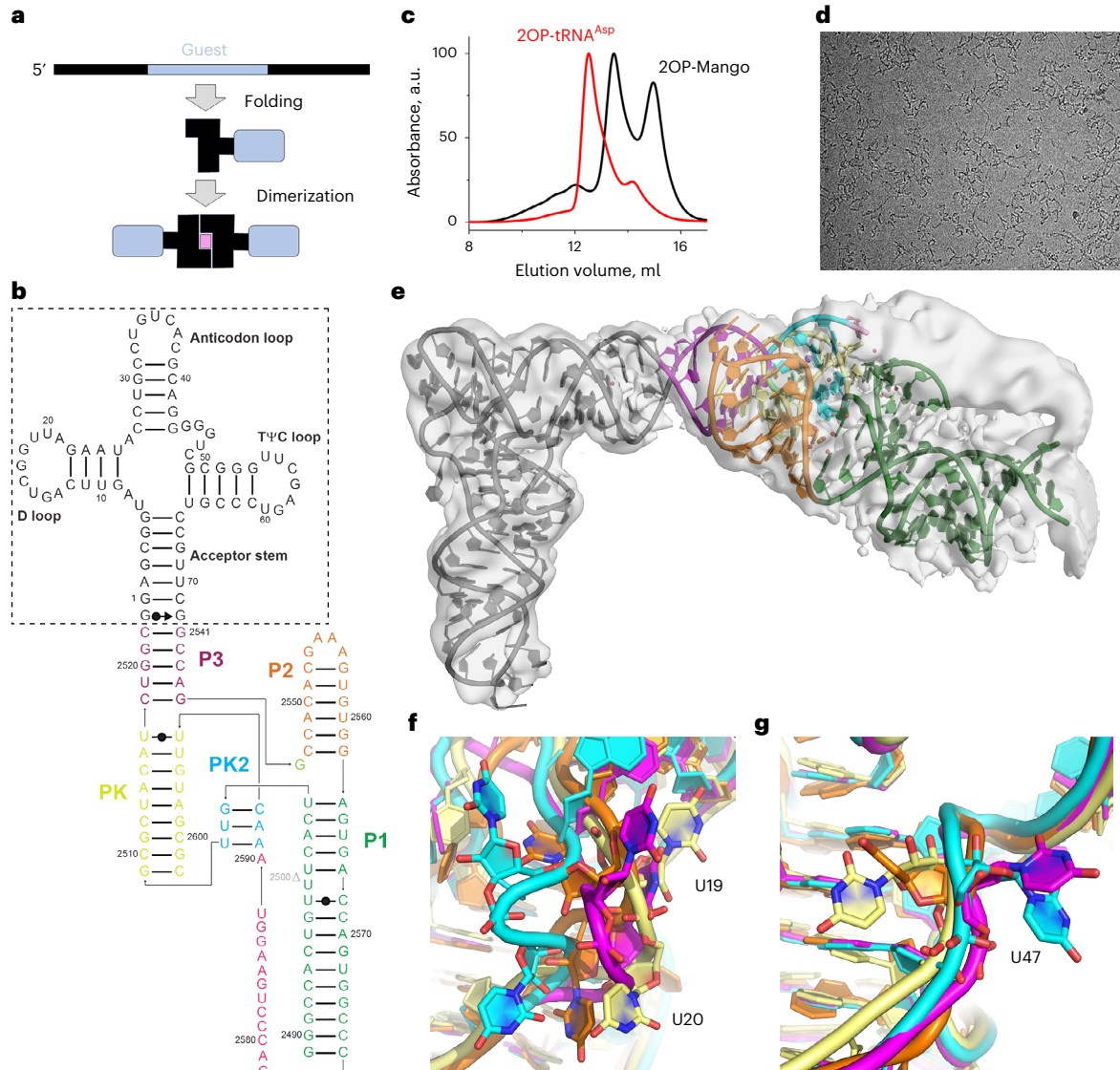

**Fig. 2 | 2OP–tRNA$^{Asp}$ structure. a**, Heuristic for designing 2OP-guest chimeras. The guest sequence (blue) is placed within the host scaffold sequence (black). Folding and dimerization via PD (pink) leads to formation of C2-symmetric particles. **b**, Secondary structure of monomeric 2OP–tRNA$^{Asp}$, colored as in Fig. 1, except the tRNA$^{Asp}$ portion is black and boxed. **c**, Size-exclusion chromatogram of folded 2OP–tRNA$^{Asp}$ (red) and 2OP–Mango (black) (absorbance at 254 nm). Traces were normalized to 100 arbitrary units (a.u.). The void volume is 8 ml. **d**, Example micrograph of 2OP–tRNA$^{Asp}$. **e**, Locally refined 2OP–tRNA$^{Asp}$ map (white surface) contoured at 10 σ in Pymol overlaid on cartoon representation of the structure. One protomer of the scaffold and tRNA$^{Asp}$ are colored as in Fig. 1, and gray, respectively. **f**, Superposition of tRNA$^{Asp}$ structures, PDB 6UGG chains A and B (cyan and magenta, respectively), tRNA$^{Asp}$– AspRS complex, PDB 1C0A (yellow) and 2OP–tRNA$^{Asp}$ (orange). Residues 19 and 20 are shown in all-atom representation for each structure. **g**, Overlay of tRNA$^{Asp}$ structures colored as in **f**, with U47 in all-atom representation.

Extended Data Fig. 1a), removing C2500 (RSV Prague C strain numbering), and mutating U2564G to create a C–G base pair. Furthermore, the transversion U2579A was introduced to convert a pyrimidine– pyrimidine pair in PD to a U–A pair. Two additional mutations in PD, C2578A and C2582U were included in optimized construct RSV1 (Extended Data Fig. 1a). Alternatively, as the core residue mutation A2546G was observed to enhance translational frameshifting in vitro by ~0.5-fold, likely by strengthening hydrogen bonding[19], this mutation was included in construct RSV2 (Fig. 1b).

Each construct (99 nt, molecular mass ~33 kDa) was screened by analytical size-exclusion chromatography (SEC) and found to be superior to the original variant in dimer yield (Fig. 1c). Optimized folding conditions were compatible with cryoEM grid preparation, and the RNAs required no further purification (Methods and Fig. 1d,e). Grid ice thickness was optimized starting from settings determined for the RSV FSE

(Methods). Consistent with chromatography, micrographs contained a majority of dimeric 'peanut'-shaped particles (Fig. 1d,e). Classification and refinement yielded C2-symmetric maps of ~3.5-3.6 Å resolution (Methods, Fig. 1f,g and Extended Data Fig. 1b,c). Of note, while the best regions of the parental RSV FSE maps[19] were the central portion including PD, RSV1 and RSV2 maps are relatively poorly resolved in these regions, which were not fully modeled (Extended Data Fig. 1d,e). The 'best to worse' resolution designations (Extended Data Fig. 1d,e and all other local resolution maps) follow recently published community recommendations for cryoEM reporting[24]. Like in the parental RSV FSE, RSV1 and RSV2 dimers are also stabilized by complementary stacking interactions in which P1 of one protomer stacks on PK2 in the other (Fig. 1a,b and Extended Data Fig. 1a). Similarly, 3D variability analyses suggested 'bending' motions normal to the major particle axis are also present among particles (Extended Data Fig. 1f,g).

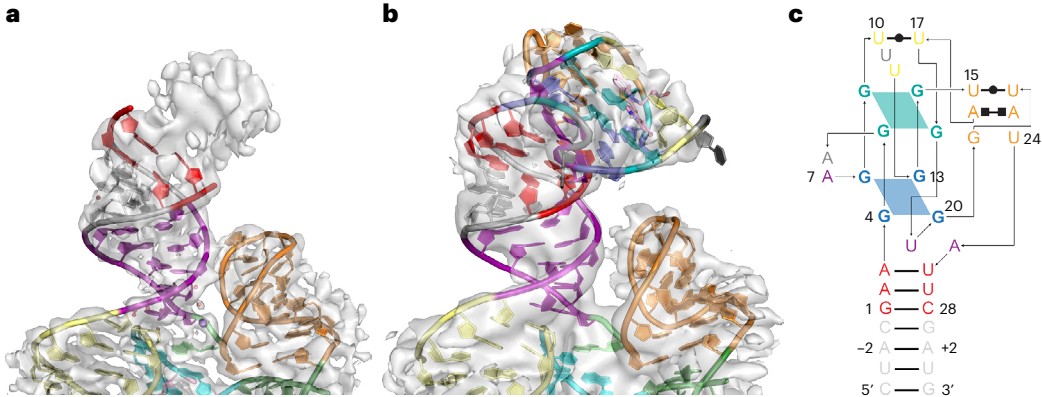

**Fig. 3 | The Mango-III aptamer is poorly ordered in the absence of ligand. a,b**, Map and model of 2OP–Mango in the absence (**a**) and presence (**b**) of its cognate fluorophore TO1-biotin. The scaffold is colored as in Fig. 1. The map thresholds are 10 σ and 15 σ for **a** and **b**, respectively, in Pymol. **c**, Secondary structure of iMango-III guest, colored as in **a** and **b**. G quartets are indicated by blue and teal diamonds.

## Application of the optimized quaternary scaffold to tRNA[Asp] and iMango-III A10U

We next used the twofold optimized RSV FSE variant RSV2 (hereafter '2OP') to design RNAs in which the apical portion of P3 was replaced with either unmodified *E. coli* tRNA[Asp] (2OP–tRNA[Asp]) or iMango-III A10U (2OP–Mango) (Fig. 2b and Extended Data Fig. 3a). In each case, host–guest chimeric RNAs were transcribed in vitro, purified under denaturing conditions, folded and concentrated before biochemical evaluation and grid preparation (Methods). Both chimeras contained nonstandard base pairs in the stem connecting the guest to the host, to serve as fiducial markers for map quality in this region (Fig. 2b and Extended Data Fig. 3a). The rationale for this was that for de novo chain tracing to be objective, maps must be of sufficient quality to discriminate between Watson–Crick and noncanonical base pairs. We expected that the fiducial marker would be particularly important when the resolution of the final map was not sufficiently high. As the mismatch may be destabilizing, it could be omitted if higher-resolution RNA reconstructions become routine. While SEC analyses showed that 2OP–tRNA[Asp] is primarily dimeric, 2OP–Mango is a mixture of oligomeric states (Fig. 2c).

Micrographs clearly showed additional density corresponding to the guests (Fig. 2d). For 2OP–tRNA[Asp], anticodon–anticodon kissing-loop interactions were occasionally present, resulting in box-like particles (Extended Data Fig. 2a). Topaz[25], pretrained on guest-free scaffolds, efficiently performed initial particle picking (Methods). The tRNA moiety of 2OP–tRNA[Asp] was relatively poorly aligned. 3D variability analysis suggested bending motions giving rise to displacements of the tRNA moiety by up to ~10–15 Å (Extended Data Fig. 2b). Local refinement of symmetry-expanded particles improved tRNA[Asp] maps to a nominal resolution of 3.7 Å (Fig. 2e and Extended Data Fig. 2c), likely by compensating for these displacements.

In the final maps, the overall tRNA shape and grooves are evident, as is base pairing in the best resolved map regions (Extended Data Fig. 2d). The least resolved regions include portions of the elbow and the anticodon loop. U19 and U20 (tRNA numbering), which are dihydrouridines in the eponymous D-loop of fully modified tRNAs, were originally found in two alternate conformations in crystals[21] (Fig. 2f). While somewhat ambiguous in our maps, the unmodified uridines in our sample seem to more closely resemble the conformation of chain A of the free tRNA structure and the RNA conformation in the structure of modified tRNA[Asp] bound to aspartyl-tRNA synthetase (AspRS)[26] (Fig. 2f). U47 adopts slightly different conformations in the crystal (Fig. 2g). In both, its nucleobase is flipped out toward a symmetry-related molecule, and its 2′-OH hydrogen bonds with a nonbridging phosphate oxygen of C50 (ref. 21). Our maps suggest that in our samples, U47 instead flips in toward A21 (Fig. 2f). Indeed, this flipped-in conformation is

more similar to that observed in the structure of the tRNA[Asp]–AspRS complex[26]. Overall, our host–guest approach produces cryoEM reconstructions of sufficient quality to delineate conformational differences between the fully post-transcriptionally modified tRNA[Asp] in previous structures and the unmodified tRNA[Asp] in our sample.

The smallest guest we examined, iMango-III A10U, is a fluorogenic aptamer[27] that binds to its cognate conditional fluorophore (TO1-biotin) using an RNA quadruplex-containing binding site[20]. As the structure of the unliganded RNA is unknown, we collected data on *apo-* and ligand-bound 2OP–Mango (Extended Data Fig. 3b,c), obtaining reconstructions at ~3.0 Å overall resolutions for both although the guest portions are poorer (Fig. 3a,b and Extended Data Fig. 3d,e). Compared to 2OP–Mango bound to TO1-biotin (Fig. 3b), the guest portion of *apo*-2OP–Mango is poorly resolved, allowing only for a portion of the connecting helix to be modeled (Fig. 3a and Extended Data Fig. 3f,g). Neither map was of sufficient quality to fully trace the intricate aptamer de novo (Fig. 3c), but the ligand-bound cryoEM map is consistent with the overall crystal structure of iMango-III A10U, which can readily be rigid-body docked (Methods). Moreover, these maps indicate that TO1-biotin fluorophore binding drastically reduces the conformational space sampled by the turn-on aptamer.

## Scaffolded U1A RNA–protein complex forms D2-symmetric tetramers

We next applied the 2OP scaffold to investigate the structure of small ribonucleoprotein complexes (RNPs). The U1 small nuclear ribonucleoprotein A (U1A) protein and cognate stem-loop are an extensively studied RNP that has previously been employed as a crystallization module[28]. Here we attached the U1A stem-loop to the 2OP scaffold (2OP–U1A) and mixed the RNA with the U1A protein before grid preparation without further purification (Extended Data Fig. 4a,b). In the resulting cryoEM reconstruction, the guest portion of 2OP–U1A was poorly resolved (and therefore not modeled), but the host portion adopted a D2-symmetric quaternary structure. At present, we do not have a full understanding of what experimental variables control C2- versus D2-symmetric oligomerization. This higher symmetry yielded averaged cryoEM maps with a resolution of 2.7 Å (Fig. 4a and Extended Data Fig. 4c,d). The refined structure shows that even though the sequence of the D2-symmetric 2OP RNA is identical to that of the C2-symmetric RNA, the RNA adopts an alternative base-pairing scheme. In the D2 structure (Fig. 4b), slippage of a C-rich tract leads to flipping out of the 5′-most C2575 and alternative pairing of subsequent C's. Rather than pairing within each dimer, C2581, C2582 and U2583 pair with A2586, G2587 and G2588 in the second dimer (Fig. 4c). C2580 perpendicularly interacts with the sugar edges of A2586 and G2587 (Fig. 4c).

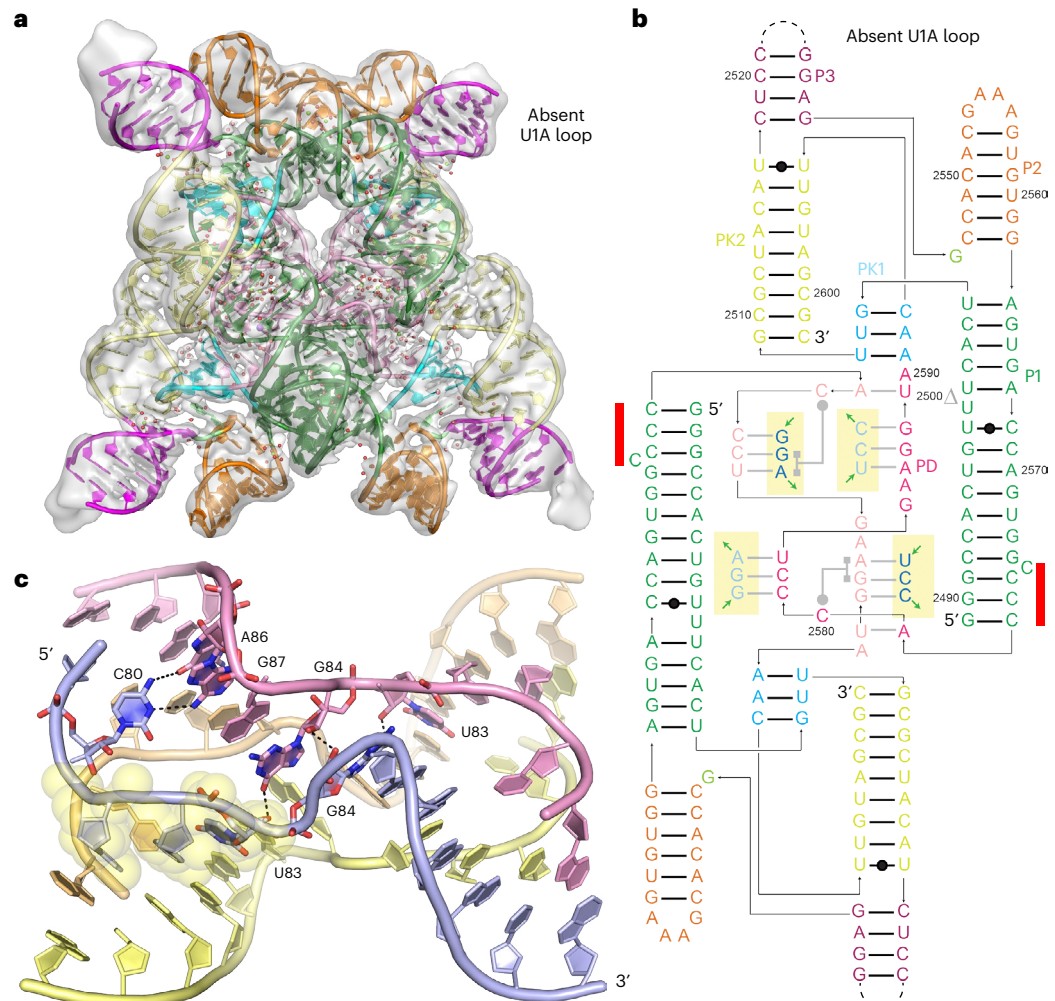

**Fig. 4 | 2OP–U1A particles form a D2-symmetric tetramer with a well-resolved core. a**, CryoEM map of D2-symmetric 2OP–U1A overlaid on cartoon representation of the structure, colored as in Fig. 1. **b**, Secondary structure of RNA shown in **a**, colored as in **a** except that PD from the second dimer are boxed in yellow and colored blue. All other residues from the second dimer are omitted for clarity. Dashed lines indicate the location of unresolved guests. Red bars indicate the C-rich tract with alternative base pairing in C2- and D2-symmetric scaffolds.

**c**, Aspect of D2-symmetric core with each PD strand colored individually, consisting of a dimer of pink and blue strands and a dimer of orange and yellow strands. Black dotted lines indicate hydrogen bonds involved in C2580·A2586[sym]·G2587[sym], U2583·G2584[sym] and G2584·G2584[sym] tertiary interactions, for which all-atom models are shown. Nucleotides 2581-3 flip out to base pair with a symmetry mate and are highlighted yellow. In **c**, numbers are abbreviated to the last two digits.

Of note, D2 tetramers were also present for *apo*-2OP–Mango but were preferentially oriented (Extended Data Fig. 5a–d), possibly due to the exposed apolar face of the G-quadruplex adhering to the air–water interface. Those particles were characterized by a top-like distribution in which the thinnest side view was absent, a sampling issue not improved by symmetry[29]. Collecting at a 45° stage tilt solved the preferred orientation problem (Extended Data Fig. 5d) but ultimately yielded maps of slightly lower quality (Extended Data Fig. 5e–h). If C2-symmetric maps had been unavailable, optimizing sample preparation for tilted data and collecting larger datasets at smaller tilt angles[30] may have improved D2-symmetric 2OP–Mango reconstructions.

### The quinine aptamer recognizes its ligand through shape complementarity

One strategy for developing RNAs with new ligand specificities is to randomize portions of a biologically evolved RNA and subject the resulting molecules to in vitro selection[22,23,31]. Recently, by applying this methodology to the aptamer domain of the bacterial adenine riboswitch, RNAs that specifically recognize the antimalarial compound quinine were discovered[22]. We fused the quinine-I aptamer sequence[22] to the

2OP scaffold (2OP–quinine) (Extended Data Fig. 6a) and processed cryoEM data for twofold symmetric particles (Extended Data Figs. 6b and 7). The resulting Coulomb potential maps were of excellent quality and extended to 2.7 Å for the locally refined host core and 2.9 Å for the locally refined quinine aptamer (Fig. 5a and Extended Data Fig. 6c–e). This allowed for de novo chain tracing of the entire quinine-I aptamer. The high quality of the map is apparent, for instance, in the portion corresponding to the noncanonical U40•C98 base pair[22] of the aptamer, which is distinctly different from those for the neighboring Watson–Crick base pairs (Extended Data Fig. 6f). By combining models for the core and aptamer, a nearly complete structure of 2OP–quinine could be built (Extended Data Fig. 6e).

The overall fold of the quinine aptamer is closely related to that of its parental riboswitch. It consists of three paired elements (P1–3) joined by loops J1/2, J2/3 and J3/1 (Fig. 5c). As in the bacterial RNA from which it was derived[22,31], the ligand-binding pocket lies at the three-way junction, the best resolved region of our quinine aptamer map (Extended Data Fig. 6g). The distal loops of the P2 and P3 helices make loop–loop interactions at the tip of the aptamer, and these interactions, which include the G23•C47 and G24•C46 base pairs buttressed

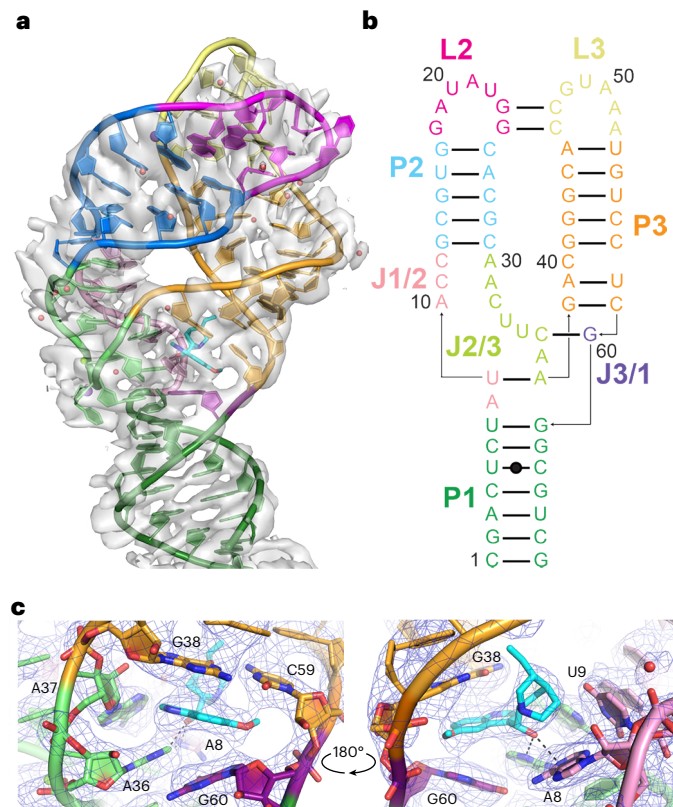

**Fig. 5 | The quinine-I aptamer uses shape complementarity to recognize its ligand. a**, Locally refined quinine aptamer CryoEM map contoured at 25 σ in Pymol, overlaid on a cartoon representation of the aptamer structure. The bound quinine molecule is indicated in cyan, and well-resolved water molecules, Mg²⁺ ions and K⁺ ions are shown as red, lime and purple spheres, respectively. **b**, Secondary structure of RNA guest in the same color scheme. **c**, View of the ligand-binding pocket from the direction of the planar quinoline ring of the ligand. Black dashed lines indicate hydrogen bonds. **d**, View from the direction of the quinuclidine ring of the ligand with hydrogen bonds between A8, A36 and the quinine hydroxyl prominent. Maps (blue mesh) in **c**,**d** are contoured at 30 σ in Pymol.

by A-minor interactions[32] from A49 and A50, are less well resolved (Extended Data Fig. 6g). The quinine binding site is organized by residues from J1/2, J2/3 and J3/1, which coalesce to organize the three-way junction. In particular, residues 30–37 in J2/3 form a consecutive series of tertiary interactions and directly contact the ligand (Fig. 5c). In the core of the aptamer, canonical and noncanonical base pairs closely complement the characteristic shape of quinine (Fig. 5d). G38•C59 stacks on top of the quinoline ring, and U9•A37, which is nearly perpendicular to G38•C59, makes van der Waals interactions with the quinuclidine moiety (Fig. 5c), which is otherwise solvent-exposed. A8 and A36 interact via a single hydrogen bond between the N1 of A8 and N6 of A36, both of which also hydrogen bond with the quinine hydroxyl group (Fig. 5d). The latter are the only two observed RNA–ligand hydrogen bonds. Finally, G60 stacks below the quinoline ring and pairs with C35, which stacks on A36. Due to A36 lying between the quinoline ring and C35•G60, the latter is not coplanar with quinoline or G38•C59 (Fig. 5c). Consistent with our structure, the mutation G60A was previously shown to be detrimental to activity of a quinine aptamer reporter in *Bacillus subtilis*[22].

## The bacterial 8-oxoguanine riboswitch envelops its ligand with hydrogen bonds

An oxidative damage product of guanine, 8-oxoG is recognized by a class of hitherto structurally uncharacterized riboswitches

in *Paenibacillaceae*[23]. We fused the 8-oxoG aptamer sequence to the 2OP scaffold (2OP–8-oxoG) for structure determination (Extended Data Fig. 7a,b). Particles adopted the tetrameric D2 symmetry, which facilitated map reconstructions extending to ~3.3 Å resolution from screening data collected on the 200 kV Glacios microscope (Extended Data Table 1), and to ~2.5 Å resolution for the host and ~2.9 Å for the 8-oxoG aptamer guest from data collected on the 300 kV Krios microscope (Fig. 6 and Extended Data Fig. 7c,d). The riboswitch aptamer RNA was readily traced in its entirety using our locally refined maps, and the entire structure of 2OP–8-oxoG was determined (Fig. 6c and Extended Data Fig. 7e). The best regions of the locally refined map of the core reveal the locations of many ordered water molecules, hydrated Mg²⁺ ions, and alternate conformers for several residues (Extended Data Fig. 7f). 3D variability analysis (Extended Data Fig. 7g) captures tilting motions between dimers in the tetrameric host core.

As predicted by homology[23], the overall structure of the 8-oxoG riboswitch aptamer domain is similar to those of the purine, deoxyguanine, and tetrahydrofolate (THF) riboswitches (Fig. 6b). As in the purine riboswitch[33] and the quinine aptamer described above, residues in J2/3 participate in numerous tertiary interactions with residues adjacent to the ligand-binding pocket and also directly contact the bound ligand (Fig. 6c,d). The best resolved region in the locally refined map (Extended Data Fig. 7h) shows how the 8-oxoG binding site is composed of stacking interactions and hydrogen bonds that interact with each of the seven hydrogen bond donors and acceptors of 8-oxoG (Fig. 6c). Similar to interactions between guanine and the purine riboswitch[33], C58 hydrogen bonds with the Watson–Crick face of 8-oxoG, U35 hydrogen bonds with the Hoogsteen edge of 8-oxoG, and the 2′-OH

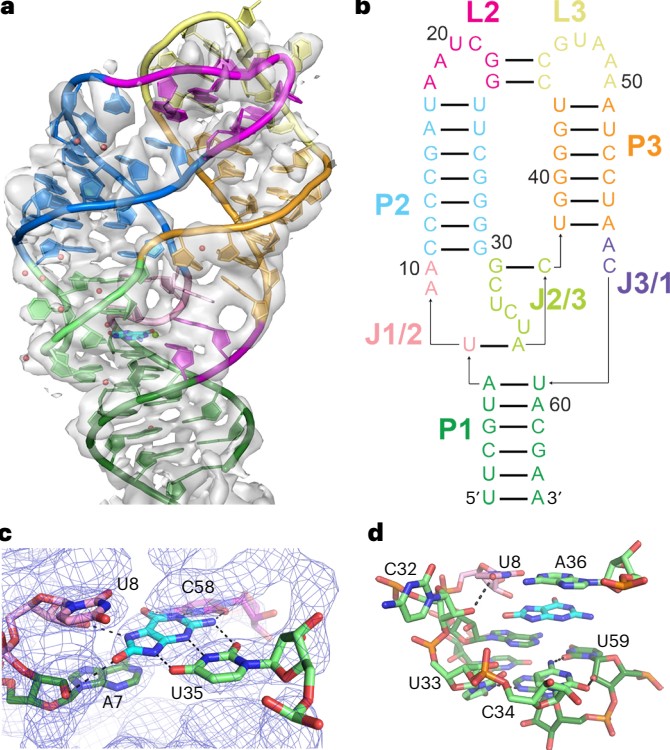

**Fig. 6 | The 8-oxoG riboswitch envelops its ligand in an intricate hydrogen bonding network. a**, Map for the 8-oxoG riboswitch aptamer domain, contoured at 30 σ in Pymol, superimposed on a cartoon representation of the structure. The bound 8-oxoG molecule is indicated in cyan and well-resolved water molecules and Mg²⁺ ions are shown as red and lime spheres, respectively. **b**, Secondary structure of the 8-oxoG aptamer guest, colored as in **a**. **c**, Binding site for 8-oxoG, with hydrogen bonds indicated by black dotted lines and map (blue mesh) contoured at 50 σ in Pymol. **d**, Tertiary interactions contributed by J2/3 residues interacting with nucleotides forming the 8-oxoG binding pocket.

of U8 hydrogen bonds with the N7 of 8-oxoG. Uniquely to the 8-oxoG aptamer, the 2′-OH of A7 hydrogen bonds with the O8 group of 8-oxoG (Fig. 6c). Nearby pyrimidines C32 and C34 form tertiary interactions with the U8•A36 and A7•U59 base pairs, which stack on either face of the bound 8-oxoG (Fig. 6d). This arrangement of tertiary interactions makes space for the O8 substituent absent in guanine, which binds to the 8-oxoG aptamer ~tenfold more weakly than 8-oxoG[23].

## Discussion

In this work, we demonstrate that an optimized RNA scaffold with intrinsic C2- or D2-symmetric quaternary structure enables de novo structure determination by single-particle cryoEM of synthetic and natural guest RNAs. This approach can be applied generally to RNAs of unknown 3D structure but with predicted or hypothesized secondary structure, which can be readily fused to our optimized host (Fig. 2a). Our experience indicates that after routine purification and folding, no further purification of the chimeric RNAs is required, expediting grid preparation and map generation. Data analysis, along with microscope time, are ultimately rate-limiting. It is possible that scaffold attachment could stabilize guests, especially if the connecting guest helix is dynamic. In the cases of the quinine aptamer and the 8-oxoG riboswitch aptamer domain, chain tracing during model building was aided by combining guest and core models (Extended Data Figs. 6e and 7e) to determine sequence registers.

Use of scaffolds for cryoEM structure determination (Supplementary Table 1) can impact, in principle, all facets of the methodology, including sample stability and folding, particle picking, alignment, and averaging. To what extent the increased molecular mass of the sample, its improved stability and homogeneity, and the symmetry of the particle result in improved final reconstructions and atomic models is difficult to deconvolute. Recent reconstructions of free tRNA[34] are of among the smallest RNA molecules investigated to date by single-particle cryoEM and were limited to 4.5-5 Å resolution. In comparison, our reconstruction of tRNA, using the twofold symmetric scaffold, has a resolution of ~3.7 Å. In our approach, C2 or D2 symmetry is applied for all final core local refinements as the maps still possess internal symmetry. In addition, symmetry has been applied for maps before local refinement during the analysis process; however, for guest RNA local refinements after symmetry expansion, symmetry is no longer applied as these maps are either no longer symmetric due to particle subtraction or application would be inappropriate, due to particle duplication (or quadrupling) during symmetry expansion. It is possible that symmetry is particularly useful for lower resolution datasets. Comparison of core refinements with and without C1 symmetry suggests that for higher-resolution datasets, each doubling of symmetry improves resolution by ~0.1 Å (Extended Data Table 2).

Scaffold application requires guests to have a helix formed by base pairing between their 5′ and 3′ ends, a frequent structural feature of compact RNAs, but one that is violated by pseudoknots, whose ends are typically spatially distant. For pseudoknots, one could envision using a permuted version of our RSV FSE-derived scaffold, joining the natural 5′ and 3′ ends (Fig. 1a), which are juxtaposed and stack on one another in dimers. Placing new 5′ and 3′ ends at P3 would thus allow for inclusion of a pseudoknot RNA at P3 or even 'sticky' ends for adding guests in *trans*. One can envision a future approach in which scaffolds and unknown RNA or DNA are mixed in *trans*, allowing for rapid screening of engineered scaffolds and guest targets. Both scaffold and target could include modifications designed with rapid downstream structure identification in mind.

As our quaternary structure scaffolds were engineered from a viral RNA genomic element that promotes frameshifting[19,35], their sequences retain codon-level information not required for function as a structure-determination tool. Moreover, dimerization is not known to be important for their viral function, and there is no connection per se between their natural function and behavior on cryoEM grids

aside from engineered oligomerization imparting sample stability. Thus, our symmetric scaffolds can in principle be further engineered in the future to be better cryoEM specimens, for example, by dampening bending motions (Extended Data Figs. 1g,h and 7g) or promoting tetramerization. Tetrameric particles ultimately yielded our best maps (Fig. 4 and Extended Data Fig. 7c); however, it is unclear at present why some host–guest complexes formed tetramers while others did not. RNA particles adopting even higher symmetry[16,36–38] may be discovered and similarly applied even if symmetry is unrelated to biological function, further enhancing the power of the approach that we have described.

## Online content

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

## Methods

Here we describe cryoEM sample preparation and evaluation, grid preparation, cryoEM data collection and analysis, map interpretation and model building. CryoEM reporting follows recently published community recommendations[24]. Maps and models are summarized in Supplementary Table 2.

### Sample preparation

RNAs were transcribed in vitro as previously described[40]. Plasmids and primers (Supplementary Table 3) were ordered from Integrated DNA Technologies, dissolved in diethylpyrocarbonate (DEPC)-treated water, and used as provided. Transcription templates were PCR-amplified using a T7 promoter forward primer and 2′-O-methylated reverse primer[41]. A self-cleaving hammerhead ribozyme was placed at the 5′ end of the RNA to ensure homogeneous ends[42]. Transcriptions were performed as previously described[40] and purified via denaturing polyacrylamide gel electrophoresis. Typical transcription reactions (5 ml) yielded 5–10 mg purified RNA, which were washed with 0.5 M KCl once and three times with DEPC-treated water, filtered with 0.1-μm Amicon ultracentrifugal filters and stored at −20 °C.

Before grid freezing, 2OP-guest RNAs were folded at 0.06 g l$^{-1}$ in RNA folding buffer (25 mM HEPES-KOH, pH 7.4 and 150 mM KCl). RNAs were heated for 3 min at 95 °C and placed on ice for >5 min. After addition of Mg$^{2+}$ acetate to 10 mM final concentration, samples were heated for 15 min at 37 °C and concentrated using 10,000 MWCO Amicon ultracentrifugal filters to ~6 g l$^{-1}$. Samples were temporarily stored on ice or at 4 °C for longer term storage. For the 2OP-Mango sample containing ligand, TO1-biotin was added before grid preparation at a 1.5:1 fluorophore:RNA molar ratio. For 2OP-quinine and 2OP-8-oxoG, ligands were included in the folding mixture and added again during concentration in tenfold molar excess.

### Size-exclusion chromatography

To assess sample homogeneity, ~25–50 μl folded RNA (~6 g l$^{-1}$) was injected at a 0.25 ml min$^{-1}$ flow rate onto a Superdex 200 10/300 GL column (Cytiva) pre-equilibrated with RNA folding buffer. RNA was excluded at a 0.75 ml min$^{-1}$ flow rate for one column volume, monitoring absorbance at 254 nm. Chromatograms were normalized and displayed with a maximum of 100 arbitrary units for comparison.

### CryoEM grid preparation

Folded RNAs (3 μl) were pipetted onto grids that were glow discharged (45–60 s) using a Pelco easiGlow glow discharger and vitrified by rapid plunging into liquid ethane using a Vitrobot (Thermo Fisher Scientific). While specific specimen conditions varied (RNA concentration and Vitrobot settings), grids were generally optimal at 4.5–6 g l$^{-1}$ RNA using Vitrobot settings of 15-s wait time, 5.5-s blot time, 4 °C and 100% humidity. 2OP-U1A grids were optimal under similar Vitrobot settings with 1 g l$^{-1}$ RNA. To ensure 100% humidity, manual humidity was turned on for ~3 s before applying sample. For carbon grids—Quantifoil R 1.2/1.3 300 mesh (EMS) or C-flat 1.2/1.3 300 mesh (Mitegen)—blot force was 10, and for gold grids—Au flat 1.2/1.3 300 mesh (Mitegen)—blot force was 16. Upon vitrification, grids were stored in liquid nitrogen until clipping using C-clips and O-rings from Thermo Fisher Scientific or Nanosoft.

### CryoEM data collection

Specimens were screened for ice thickness and particle density to assess data quality using a Glacios microscope (Thermo Fisher) equipped with a Falcon 3 camera. Example data (Extended Data Table 1; 2OP-8-oxoG-Glacios) illustrates a common data collection strategy inserting a 100-μm objective aperture and collecting at ×150,000 magnification (0.91 Å pixel size) with an ~40–50 electron/Å$^2$ total dose over 40 frames. Data treatment followed a similar strategy to that outlined below for Krios microscope data (Supplementary Figs. 1 and 2)

although late-stage refinement routines, such as local refinement and 3D classification in cryoSPARC, were omitted for expediency. Specimens collected on gold grids were initially screened on carbon grids (for example, C Flats, 1.2/1.3 300 Mesh).

Data were collected using three different Krios microscopes located at on the National Institutes of Health (NIH) main campus in Bethesda in Building 13 or Building 37 or at Rocky Mountain Laboratory (RML) campus in Hamilton, Montana (Extended Data Table 1). For the latter, prescreened grids stored in autogrid boxes were shipped in autogrid box pucks at liquid nitrogen temperature in dry shipper dewars (Mitegen). Before shipment, autogrid boxes and pucks were cooled and manipulated using foam dewars (Mitegen) and tongs.

Krios data collection at RML used correlated-double sampling mode[43], and all other Krios data collection used super-resolution mode. For the latter, the reported pixel sizes in Extended Data Table 1 are twofold binned (Supplementary Figs. 1 and 2). Data collection for RSV1, RSV2, 2OP-tRNA$^{Asp}$, 2OP-Mango and 2OP-U1A used larger doses and wider defocus ranges on copper Quantifoil grids (R1.2/1.3 300 mesh) (Extended Data Table 1). Data collection for other samples used gold grids (AU Flats, 1.2/1.3 300 mesh) and lower doses and narrower defocus ranges. For samples on gold grids, an energy filter was used during grid square mapping and set to a zero loss point of 20 eV with a slit width of 15 eV in SerialEM[44]. This mapping mode[45] reduces high contrast from gold, allowing more easy discrimination of thin ice from empty holes and thicker ice. Consequently, a higher proportion of useful micrographs were collected.

### CryoEM data analysis

The overall data treatment pipeline used RELION[46] and cryoSPARC[47] (Supplementary Figs. 1 and 2). Motion correction used MotionCor2 (ref. [48]) in RELION binning super-resolution videos twofold when relevant, and contrast transfer function (CTF) estimation used CTFFIND4 (ref. [49]) in RELION. Micrographs were then curated to remove icy and poorly fit micrographs, and the numbers of total and curated micrographs are reported (Extended Data Table 1). For previously uncharacterized samples (from Glacios screening), particle picking used Topaz[25] trained with guest-free 2OP scaffolds. New Topaz models were later trained for subsequent picking of Krios datasets. The number of initial particle images reported in Extended Data Table 1 is the first particle set extracted from micrographs after picking, applying a figure of merit cutoff of −3. After a round of two-dimensional (2D) classification to remove ice and noise, threefold downsampled particles were imported into cryoSPARC for iterative rounds of ab initio model reconstruction and heterogeneous refinement. Typical classes consisted of a noisy class, a monomeric class and a dimeric or tetrameric class (Supplementary Figs. 1 and 2). The best particle classes were homogeneously refined, applying symmetry. These particles were imported into RELION, re-extracted without downsampling, and imported into cryoSPARC to assess resolution using homogeneous and nonuniform refinement routines. At this stage, additional sieves to improve particle classes were performed using ab initio model reconstruction and heterogeneous refinements and 2D classification. To identify the source of heterogeneity, 3D variability analysis was conducted using a filter resolution lower than the resolution of the data (for example, 5 Å). Particles were then imported into RELION for particle polishing and iterative CTF refinement. Typically, CTF refinement improved most from beamtilt refinement, then anisotropic magnification, and least from per-particle defocus. Particles were repolished and imported into cryoSPARC for final rounds of whole-map resolution assessment (Supplementary Figs. 1 and 2). 3D variability analysis was performed at this stage to assess motions present among final particles. (Before returning to cryoSPARC, it is also possible to perform symmetry expansion and particle subtraction on re-extracted polished particles, recentering on the guest and using a significantly smaller box size; however, in our experience, this did not improve map reconstructions, although

it did significantly speed up local refinement and 3D classification routines below.)

Local masks of the core and guest were made using Segger in Chimera[50]. These were low-pass filtered by 5–10 Å and expanded with 15–20 Å of soft padding in cryoSPARC (Supplementary Figs. 1 and 2). Masks for 2OP–tRNA$^{Asp}$ and 2OP–Mango enveloped the aptamer and a portion of the scaffold core, which decreases the nominal resolution of the entire map (the best map regions are in the core). Masks for 2OP–quinine and 2OP–8-oxoG enveloped only the aptamer or core. For the latter, particle subtraction used these masks to produce either (1) guest-subtracted core particles for local refinement of the core or (2) symmetry-expanded, core-subtracted particles for local refinement of the guest (Supplementary Figs. 1 and 2). Subsequent 3D classification of locally refined particles was performed before final local refinement of the best particle subset. As automatically sharpened maps in cryoSPARC were often over-sharpened, locally filtered (locally sharpened) maps were calculated using a B-factor half of that estimated by cryoSPARC (for example, for a predicted B-factor of 100 Å$^{-2}$, maps were sharpened by 50 Å$^{-2}$).

### Tilted cryoEM data analysis

For 2OP–Mango-TO1-biotin tetramers, collected at a 45° stage tilt, analysis was entirely performed in cryoSPARC, beginning with patch motion correction and patch CTF estimation. The latter resulted in better fit CTFs than in RELION, which fit CTFs to ~10 Å on average. The fraction of curated micrographs was lower in this dataset compared to others in part due to the inclusion of a greater fraction of bad ice and empty holes. Particles were picked using templates created from 2OP–U1A tetramers, and subsequent data analysis followed the same routine as above for untilted data, excluding particle polishing and CTF refinement steps. The particle viewing direction distributions from cryoSPARC (Extended Data Fig. 5c,d) were used to assess the success of tilting in increasing sampled particle orientations, as observed recently[30].

### Map interpretation and model building

While the best maps for model building in Coot[51] were locally filtered maps created in cryoSPARC, DeepEM enhanced maps[52], unsharpened maps, and automatically sharpened maps from cryoSPARC were also consulted to aid in model interpretation. Maps and models are summarized in Supplementary Table 2. For RSV1 and RSV2, the RSV FSE cryoEM model[19] was used as a starting model, incorporating mutations and deletions to match the scaffold sequences and removing poorly resolved residues in PD (Extended Data Fig. 1d,e). After one round of rigid-body refinement, models were refined in Phenix[53] using real-space refinement[54] with secondary structure restraints, noncrystallographic symmetry (NCS) restraints, simulated annealing and B-factor refinement. During later refinements, water molecules, ions, and ligands were modeled, and models were refined used similar routines without simulated annealing. As GAAA and UUCG tetraloops could not be modeled de novo, references were used for each from the SARS-CoV-2 frameshifting pseudoknot[55] and a fragment of the group I intron[56], respectively.

For *E. coli* tRNA$^{Asp}$ (Fig. 2) and Mango-III A10U (Fig. 3), previously published crystal structures[20,21] were used to initiate model building. For tRNA$^{Asp}$, whose crystallographic asymmetric unit contained two RNA chains with different conformations, each chain was aligned to the cryoEM map. Chain A from the crystal structure was used for further cryoEM model building, which implemented real-space refinement and B-factor refinement with secondary structure restraints. In addition, the 2.4 Å resolution crystal structure of *E. coli* tRNA$^{Asp}$ bound to ApsRS[26] was also used for comparison (Fig. 2). The model for *apo*-2OP–Mango was built using a monomer of 2OP as the starting model, imposing secondary structure restraints and implementing simulated annealing, real-space refinement and B-factor refinement. A potassium ion and waters were built, and subsequent refinements

omitted simulated annealing. To model 2OP–Mango with TO1-biotin, the model for *apo*-2OP–Mango was rigid-body docked and refined as above. The crystal structure of Mango-III was then rigid-body docked into the map, A10 was changed to U, and residues connecting scaffold and guest were built and real-space refined. All real-space refinement performed with the guest moiety for this dataset was restrained using the Mango-III wild-type (WT) RNA[20] as a reference model.

For the initial model of the D2-symmetric core in 2OP–U1A (Fig. 4), one copy of the C2-symmetric model for RSV2 was placed four times and rigid-body docked. Residues were rebuilt and refined iteratively, focusing on changes present within the core, after which hydrated Mg$^{2+}$ ions were placed, and individual waters were built. The criteria for building waters were that peaks be observed at 1.2 σ (in Coot) in the filtered map with distances of ~2.7–3.5 Å from hydrogen bond acceptors or donors. The criteria for building hydrated Mg$^{2+}$ were either direct coordination ~2.1 Å away from an oxygen ligand and density surrounding the Mg$^{2+}$ for waters or features too large to be waters ~4–5 Å away from ligands. Weaker unmodeled Mg$^{2+}$ are likely present in the less resolved regions of the maps, as there are unmodeled ion-like features too far away to be waters (~4.5–5.0 Å). The conformation of C2575 was initially ambiguous for 2OP–U1A, and it was only after 2OP–8-oxoG data were collected that this residue's position was confirmed.

For the C2-symmetric core of 2OP–quinine, the model of RSV2 was used as a starting point. Refinement was performed as above for 2OP, imposing NCS and secondary structure restraints. Except for four residues in PD, the entire core was modeled (Extended Data Fig. 6e). Late-stage refinements omitted simulated annealing and focused on building waters. Two strong features were modeled as K$^+$ ions.

The 2OP–U1A model was used to initially model the D2-symmetric core of 2OP–8-oxoG; however, the higher-resolution data provided greater clarity to the tetrameric interface formed by PD strands (Fig. 4c). Flipped out C2575 was evident and modeled, allowing the sequence register of PD to be traced. Alternative conformers of residues U141, G142, A143 and A144 in PD were then built (Extended Data Fig. 7f). The bases of U141 and A144 are largely unchanged, but the conformers of their phosphate backbones vary. In contrast, the nucleobases and riboses of G142 conformers are flipped in opposite directions, and the nucleobases of conformers of A143 point in the same direction but shift ~2.5 Å. The numerous waters and ions observed in the core were modeled using a similar strategy to 2OP–U1A.

For the 2OP–quinine and 2OP–8-oxoG aptamers (Figs. 5 and 6), filtered maps from local refinements of each aptamer were inspected as refinements proceeded. Once interpretable maps were obtained, the chains were traced de novo in Coot by first placing an idealized double-stranded A-form helix into density for the paired region bridging the aptamer and core. Additional residues were built one by one, assisted by real-space refinement in Coot. Real-space refinement in Phenix was conducted as above, initially with secondary structure restraints and simulated annealing and later omitting simulated annealing. Once residues surrounding the ligand-binding pocket had been traced and refined, the ligand was placed and oriented, as the ligand-binding pockets were the best resolved features of each map (Extended Data Figs. 6g and 7h). In contrast, the peripheries of these maps were of lower quality.

Owing to the excellent map quality of the quinine and 8-oxoG data, the entire RNAs (Extended Data Figs. 6e and 7e) were modeled by rigid-body docking the separately modeled aptamer and host into the final nonuniform refinement locally filtered map (at nominal resolutions of ~2.9 Å and ~2.7 Å for 2OP–quinine and 2OP–8-oxoG, respectively). While the cores of these maps are still of excellent quality, the guest portions are lower resolution than their locally refined counterparts. Missing residues connecting the guests to the core were traced, and the entire RNAs were real-space refined, using the cores as reference models. Ions and waters were not modeled in these maps, as high-resolution features are better visualized in locally refined maps.

## Reporting summary

Further information on research design is available in the Nature Portfolio Reporting Summary linked to this article.

## Data availability

Maps were deposited to the Electron Microscopy Data Bank following recent guidelines[24], and atomic coordinates were deposited in the Protein Data Bank (accession codes are provided in Supplementary Table 2). Videos for each dataset were deposited in EMPIAR. These include micrographs for RSV1 (EMPIAR-12578/data/Op1), RSV2 (EMPIAR-12578/data/Op2), 2OP–tRNA^Asp (EMPIAR-12578/data/tRNA), 2OP–Mango (no ligand) (EMPIAR-12578/data/Mango/NoLigand), 2OP–Mango (with ligand) (EMPIAR-12578/data/Mango/Ligand), 2OP–U1A (EMPIAR-12578/data/U1A), 2OP–Mango (no ligand) collected at a 45° stage tilt (EMPIAR-12578/data/Mango/Tilted), 2OP–quinine (EMPIAR-12578/data/Quinine), 2OP–8oxoG (EMPIAR-12578/data/8oxoG) and 2OP-8-oxoG-Glacios (EMPIAR-12578/data/8oxoG-Glacios).

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

## Acknowledgements

We thank M. Banco, N. Demeshkina, T. Dou, Q. Elghondakly, L. Passalacqua, S. Shaffer, H. Shamroukh and E. Zhar for discussions; E. Fischer, F. Hoyt and C. Schwartz of the NIAID Research Technologies Branch Electron Microscopy Unit for cryoEM data collection (Montana Krios) at RML; R. Huang, A.J. Morton and Z. Lang for cryoEM data collection (Building 37 Krios), which used the NCI/NICE CryoEM Facility; H. Wang for help with cryoEM data collection (Building 13 Krios) and U. Baxa for help with cryoEM screening (Building 50 Glacios), which used the NIH Multi-Institute CryoEM Facility; and Z. Yu and A. Roll-Mecak for help with cryoEM screening (Building 37 Glacios), which used the cryoEM core facility of the National Institute of Neurological Disorders and Stroke. This work used the computational resources of the NIH HPC Biowulf cluster (https://hpc.nih.gov). C.P.J. is the recipient of NIH transition award K22HL139920. This research was supported by the Intramural Research Program of the NIH. The contributions of the NIH authors were made as part of their official duties as NIH federal employees, are in compliance with agency policy requirements and are considered Works of the United States Government; however, the findings and conclusions presented in this paper are those of the authors and do not necessarily reflect the views of the NIH or the US Department of Health and Human Services.

## Author contributions

C.P.J. performed the experiments and analyzed data, and C.P.J. and A.R.F.-D. wrote the manuscript.

## Competing interests

The authors declare no competing interests.

## Additional information

**Extended data** is available for this paper at https://doi.org/10.1038/s41592-026-03016-x.

**Correspondence and requests for materials** should be addressed to Christopher P. Jones or Adrian R. Ferré-D'Amaré.

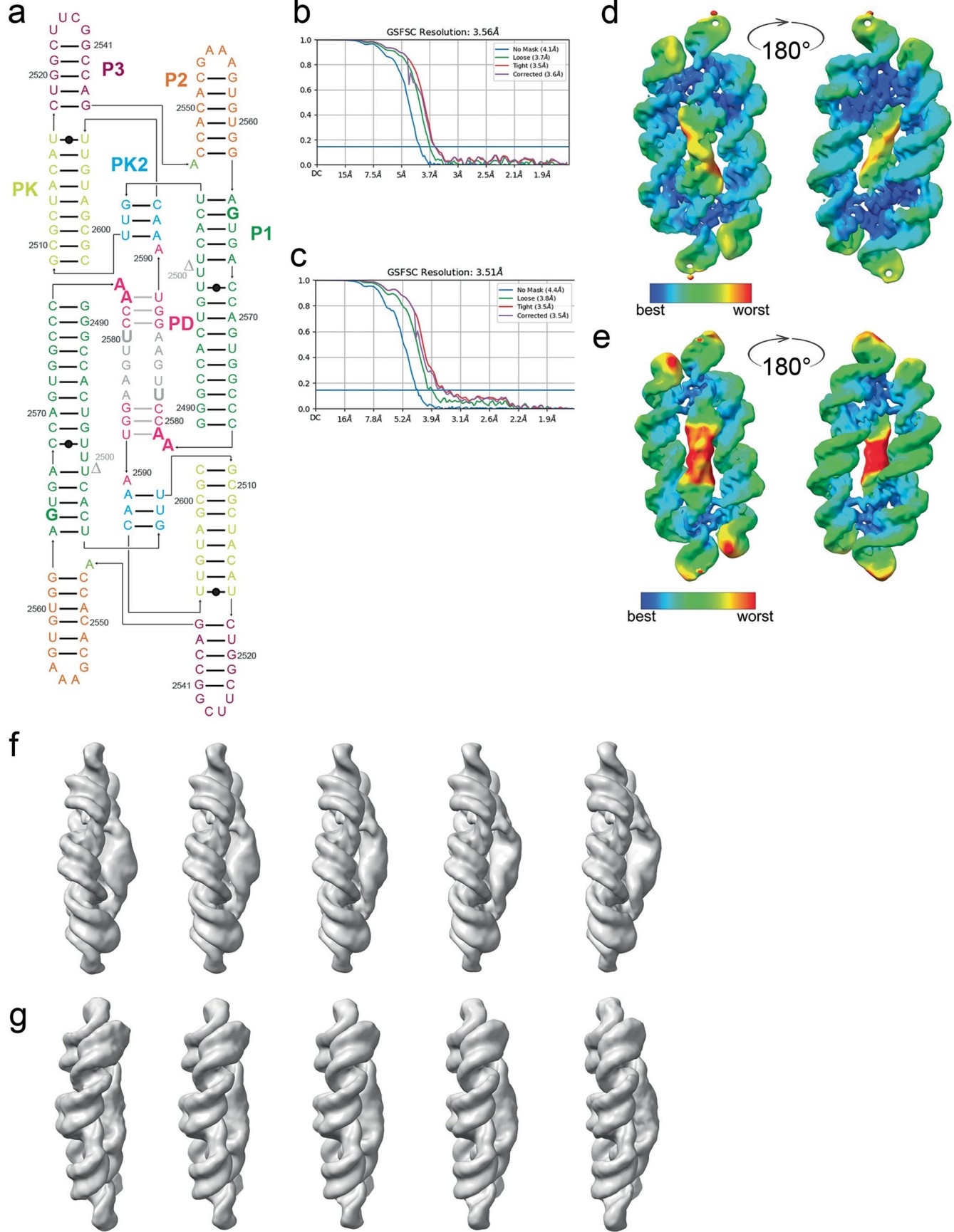

**Extended Data Fig. 1 | See next page for caption.**

**Extended Data Fig. 1 | 2OP scaffold design and optimization. a**, Secondary structure of dimeric RNA RSV1. Mutations are indicated by bold letters; gray letters denote unmodeled residues. A 'Δ' indicates the deletion of C2500. **b** and **c**, Fourier shell correlation plots for RSV1 (**b**) and RSV2 (**c**), using FSC = 0.143 as the nominal resolution criterion. **d** and **e**, Local resolution estimation for RSV1 (**d**) and RSV2 (**e**). Both maps are contoured at 0.15 map threshold in Chimera, the nominal resolutions (3.5 Å) are colored cyan, and the scales are from 3 Å (blue) to 5 Å (red). The 'best to worse' resolution designations follow recently published community recommendations for cryoEM reporting[1]. **f** and **g**, 3D variability analysis for RSV1 (**f**) and RSV2 (**g**) showing each of 5 frames for the bending mode for particles normal to the major particle axis.

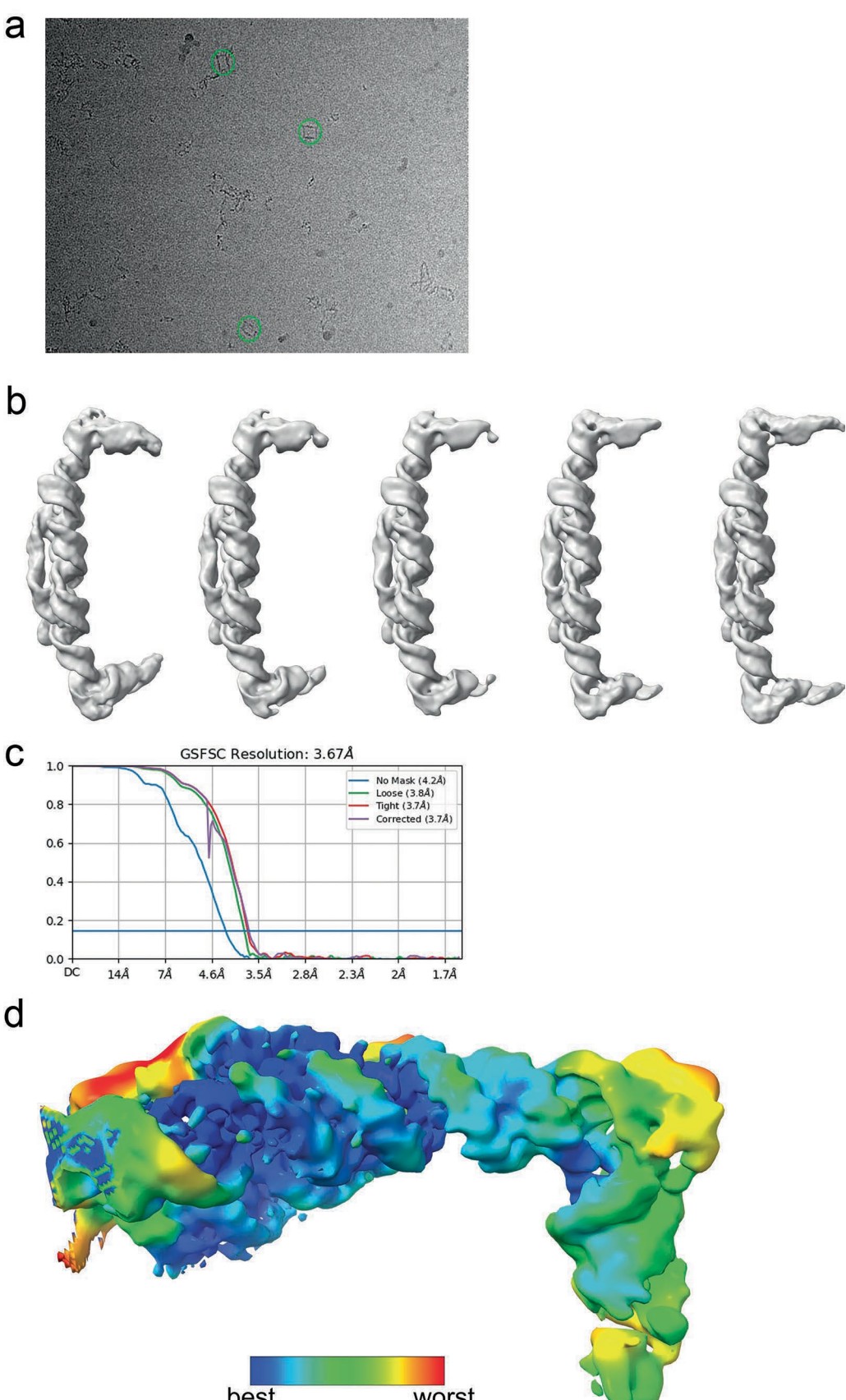

**Extended Data Fig. 2 | 2OP–tRNA^Asp design and characterization. a**, Example micrograph showing 'box-like' 2OP–tRNA^Asp particles (green circles). **b**, 3D variability analysis of 2OP–tRNA^Asp, showing each of 5 frames for the bending mode of 2OP–tRNA^Asp particles normal to the major axis. The first frame contains the most bent particle set, and the last frame contains the least bent particle set. **c**, Nominal resolution estimate for 2OP–tRNA^Asp. **d**, Local resolution estimates for best and worst map regions. The map threshold is 0.12, the nominal resolution (3.7 Å) is colored cyan, and the scale is from 3.3 Å (blue) to 6 Å (red).

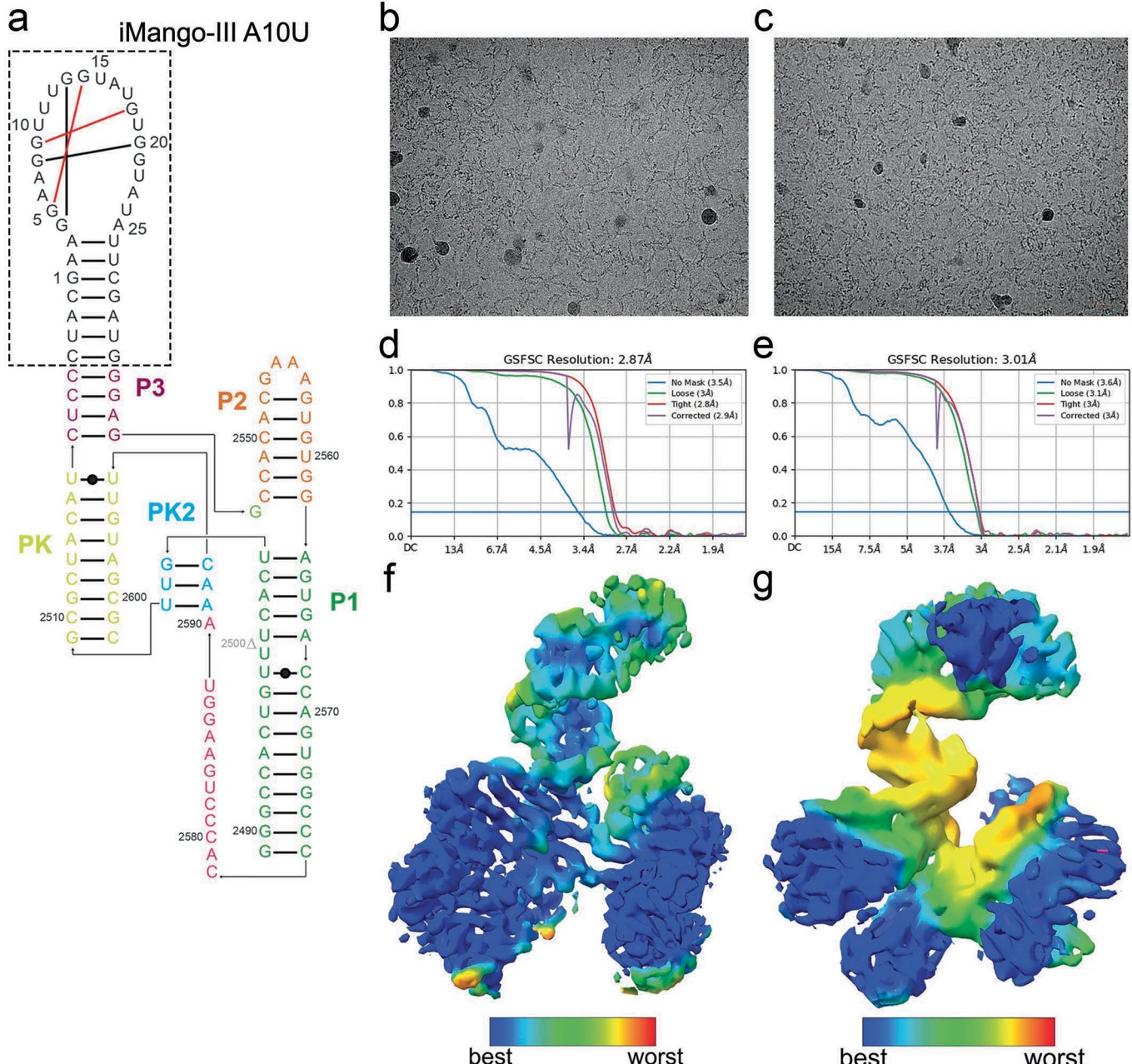

**Extended Data Fig. 3 | 2OP–Mango ligand-free and ligand-bound states.**
**a**, Secondary structure of monomeric 2OP–Mango. The iMango-III-A10U portion of the RNA is black and boxed, and black and red lines indicate pairing among G nucleobases in the first and second G quartet, respectively. The Mango portion of the scaffold is numbered as previously[2]. **b** and **c**, Example micrographs for 2OP–Mango without (**b**) and with TO1-biotin (**c**). **d** and **e**, Nominal resolution estimates for 2OP–Mango in the absence (**d**) and presence (**e**) of TO1-biotin. **f**, Local resolution estimate of best and worst map regions of 2OP–Mango in the absence of ligand. The map threshold is 0.2, the nominal map resolution (2.9 Å) is colored cyan, and the scale is from 2.6 Å (blue) to 5 Å (red). **g**, Local resolution estimate of best and worst map regions of 2OP–Mango in the presence of TO1-biotin. The map threshold is 0.2, the nominal map resolution (3.0 Å) is colored cyan, and the scale is from 2.7 Å (blue) to 5 Å (red).

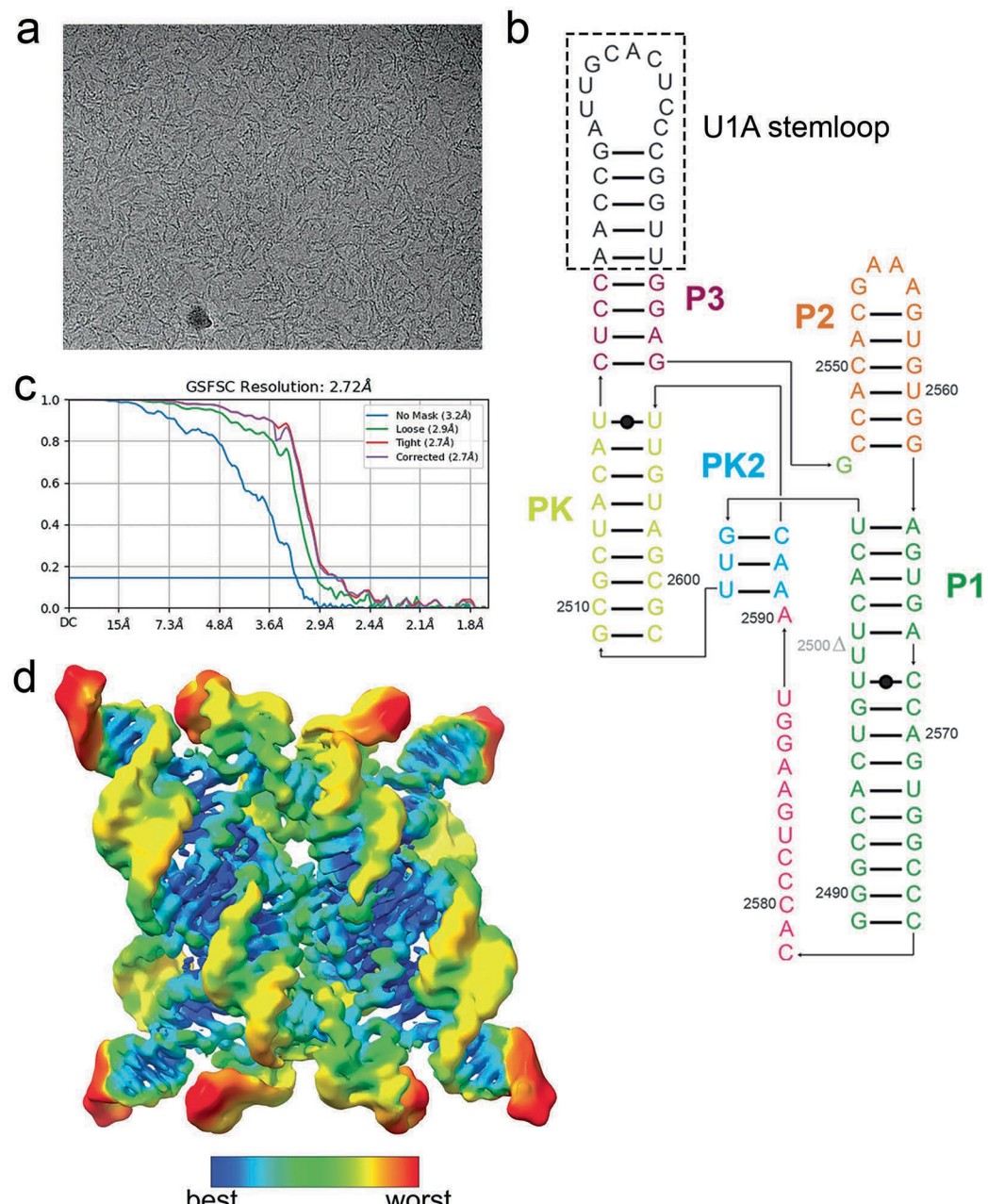

**Extended Data Fig. 4 | 2OP-U1A has a well-resolved D2-symmetric core.**
**a**, Example micrograph of 2OP-U1A particles. **b**, Sequence of 2OP-U1A. The scaffold is colored as in Fig. 1 except for the U1A stem-loop guest, which is colored black and boxed. **c**, Nominal resolution of 2OP-U1A. **d**, Local resolution estimation for 2OP-U1A. The map threshold is 0.2 in Chimera, and the nominal resolution (2.7 Å) is colored cyan, and the scale is from 2.4 Å (blue) to 4.5 Å (red).

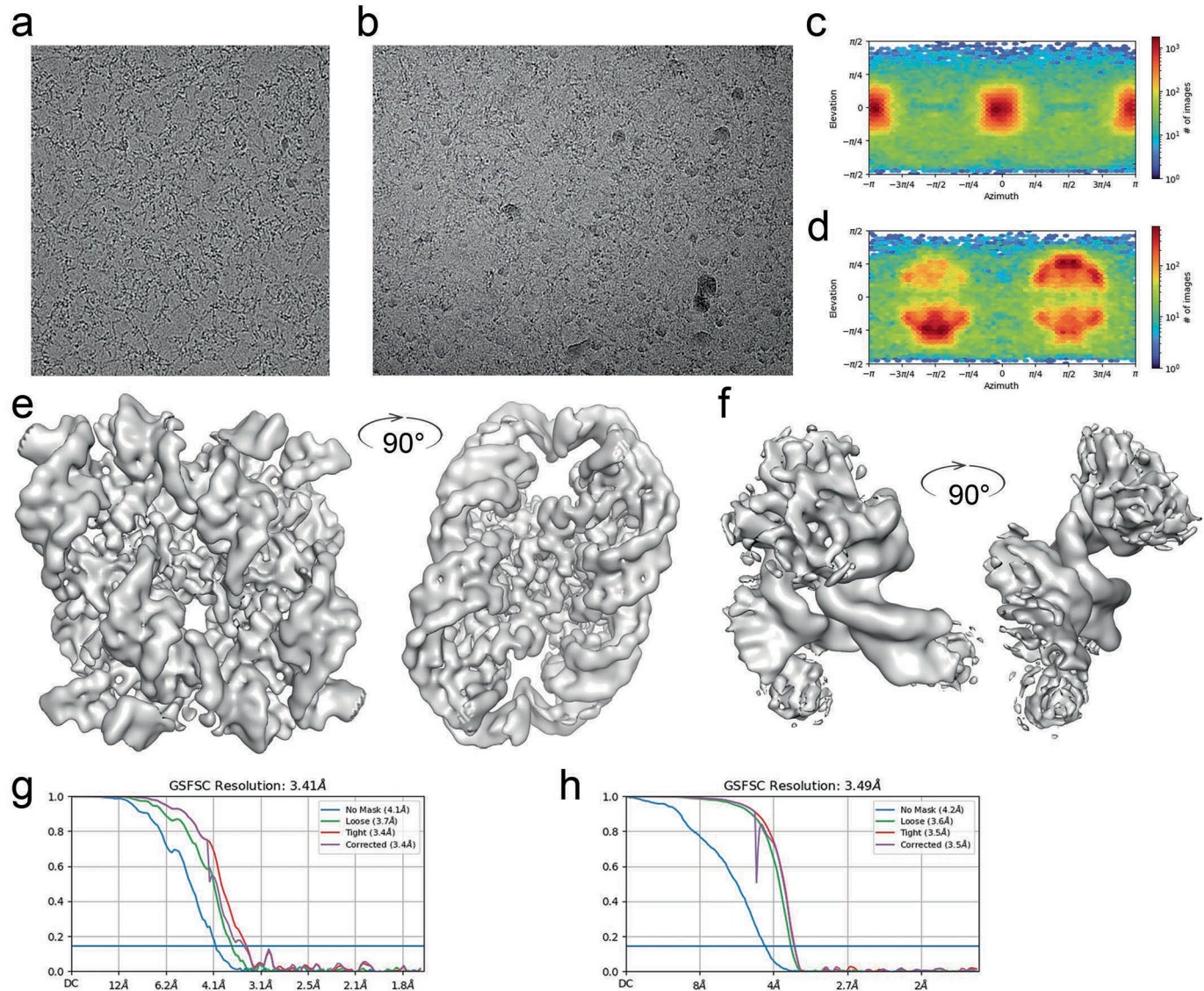

**Extended Data Fig. 5 | D2-symmetric 2OP–Mango particles are characterized by a preferred orientation solved by stage tilting. a** and **b**, Example micrographs of 2OP–Mango D2-symmetric particles, untilted during Glacios screening (**a**) and with 45° stage tilt during Krios data collection (**b**). **c** and **d**, Particle orientation of D2-symmetric particles without (**c**) and with 45° stage tilt (**d**). **e**, Views of locally refined core map from tilted data, contoured at 8 σ in Pymol. **f**, Views of locally refined aptamer map from tilted data, contoured at 10 σ in Pymol. **g** and **h**, Nominal resolution for core (**g**) and guest (**h**) maps, corresponding to **e** and **f**, respectively.

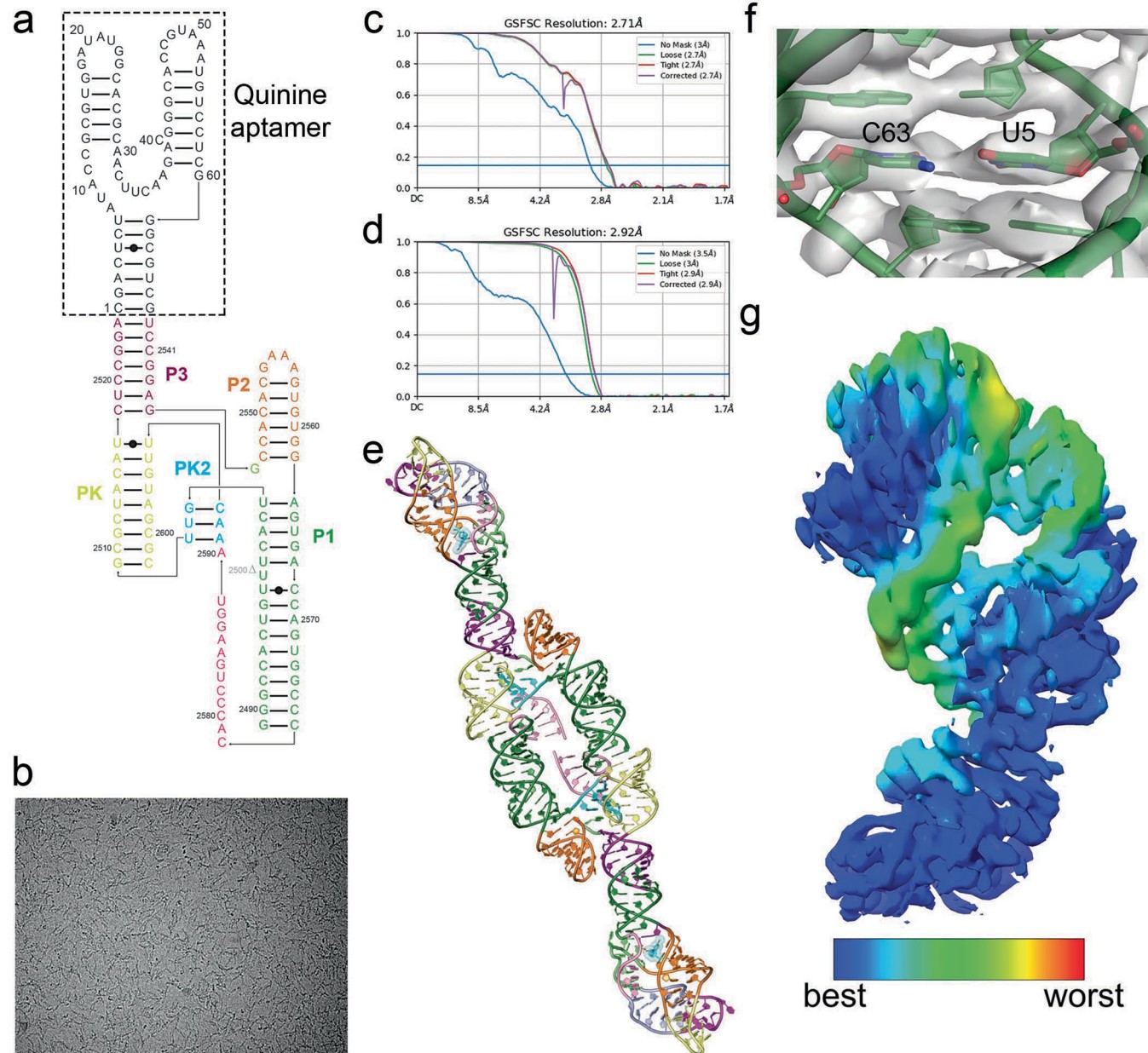

**Extended Data Fig. 6 | 2OP-Quinine guest and core characterization.**
**a**, Secondary structure of 2OP-Quinine. The core is colored as in Fig. 1, and the guest is colored black and boxed. **b**, Example micrograph of 2OP-Quinine particles. **c** and **d**, Nominal map resolutions for core (**c**) and guest (**d**).

**e**, Model of the entire 2OP-Quinine RNA. **f**, Map quality at U5·C63 pair overlaid with the cartoon model. The map threshold is 25. **g**, Local resolution estimation for guest map, contoured at a threshold of 0.25. The nominal resolution (2.9 Å) is colored cyan, and the scale is from 2.6 Å (blue) to 4.5 Å (red).

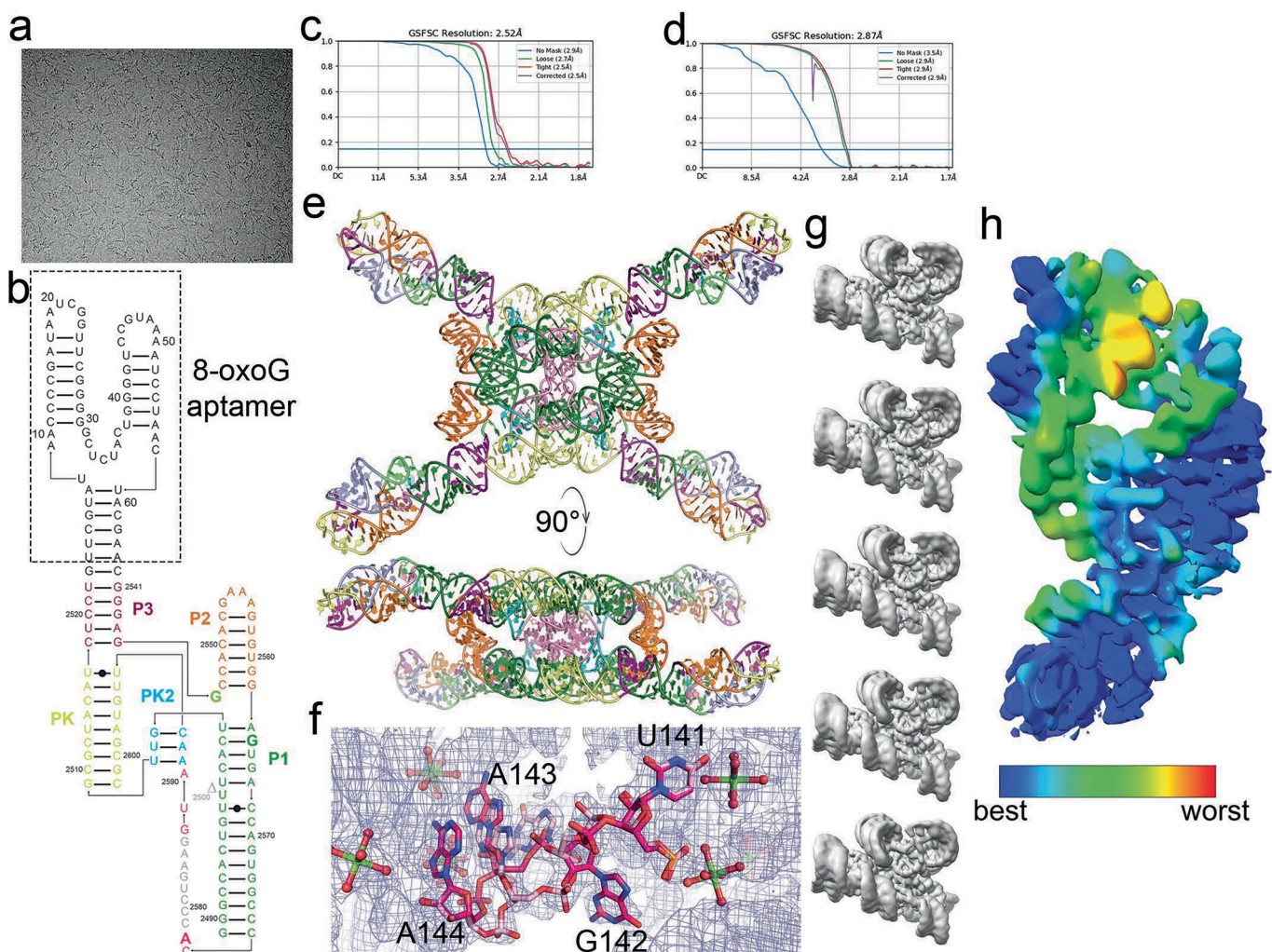

**Extended Data Fig. 7 | 2OP-8oxoG guest and core characterization. a**, Example micrograph of 2OP-8oxoG particles. **b**, Secondary structure of 2OP-8oxoG fusion. The core is colored as in Fig. 1, and the 8-oxoG aptamer guest is black and boxed. **c** and **d**, Nominal resolutions for core (**f**) and guest (**g**). **e**, Model of the entire 2OP-8oxoG RNA. **f**, Aspect of hydrated magnesium ions and alternate conformers in the core. Alternate conformers are light and dark pink and shown only for one of the four chains. Map is shown as blue mesh and contoured at 10 sigma. **g**, 3D variability analysis of the core of 2OP-8oxoG, showing two views. **h**, Local resolution estimation for 2OP-8oxoG, contoured at 0.25 map threshold. The nominal resolution (2.9 Å) is colored cyan, and the scale is from 2.6 Å (blue) to 4.5 Å (red).

**Extended Data Table 1 | Summary of cryoEM statistics**

| Dataset | RSV1 | RSV2 | 2OP-tRNA$^{Asp}$ | 2OP-Mango | 2OP-Mango (+TO1-biotin) | 2OP-U1A | 2OP-Mango (tilted) Core | Guest | 2OP-Quinine Core | Guest | 2OP-8oxoG Core | Guest | 2OP-8oxoG-Glacios Core | Guest |
|---|---|---|---|---|---|---|---|---|---|---|---|---|---|---|
| **Data collection** | | | | | | | | | | | | | | |
| Source location | NICE (Bldg 37) | RML (Montana) | NICE (Bldg 37) | RML (Montana) | MICEF (Bldg 13) | RML (Montana) | MICEF (Bldg 13) | | MICEF (Bldg 13) | | MICEF (Bldg 13) | | MICEF (Bldg 50) | |
| Microscope | Krios G1 | Krios G1 | Krios G1 | Krios G1 | Krios G3 | Krios G1 | Krios G3 | | Krios G3 | | Krios G3 | | Glacios 1 | |
| Detector | Gatan BioQuantum K3 camera | Gatan BioQuantum K3 camera | Gatan BioQuantum K3 camera | Gatan BioQuantum K3 camera | Gatan BioQuantum K3 camera | Gatan BioQuantum K3 camera | Gatan BioQuantum K3 camera | | Gatan BioQuantum K3 camera | | Gatan BioQuantum K3 camera | | Falcon 3 | |
| Date | 2023-02-10 | 2023-01-11 | 2023-02-09 | 2023-06-14 | 2024-04-04 | 2023-06-17 | 2024-05-05 | | 2024-02-29 | | 2024-04-29 | | 2024-04-26 | |
| Grid type | Quantifoil R1.2/1.3 300 Mesh | Quantifoil R1.2/1.3 300 Mesh | Quantifoil R1.2/1.3 300 Mesh | Quantifoil R1.2/1.3 300 Mesh | AU Flat 1.2/1.3 300 mesh | Quantifoil R1.2/1.3 300 Mesh | | | AU Flat 1.2/1.3 300 mesh | | AU Flat 1.2/1.3 300 mesh | | AU Flat 1.2/1.3 300 mesh | |
| Magnification | 105,000 | 105,000 | 105,000 | 105,000 | 105,000 | 105,000 | 105,000 | | 105,000 | | 105,000 | | 150,000 | |
| Voltage (kV) | 300 | 300 | 300 | 300 | 300 | 300 | 300 | | 300 | | 300 | | 200 | |
| Electron exposure (e/Å$^2$) | 54.4 | 60 | 54.4 | 60 | 39.07 | 60 | 39.15 | | 45.09 | | 39.15 | | 39.15 | |
| Frames (no.) | 50 | 60 | 50 | 60 | 30 | 60 | 30 | | 34 | | 30 | | 40 | |
| Defocus range (μm) | -1.2-3.0 | -0.5-2.5 | -1.2-3.0 | -0.5-2.5 | -0.6-1.6 | -0.5-2.5 | -0.6-1.6 | | -0.8-2.0 | | -0.6-1.8 | | -0.8-2.0 | |
| Pixel size (Å) | 0.83 | 0.864 | 0.83 | 0.864 | 0.83 | 0.864 | 0.83 | | 0.83 | | 0.83 | | 0.91 | |
| Symmetry imposed$^†$ | C2$^†$ | C2$^†$ | C2 | C2 | C2 | D2$^†$ | D2$^†$ | | C2$^†$ | | D2$^†$ | | D2$^†$ | |
| Initial micrographs (no.) | 6,741 | 5,179 | 3,932 | 6,553 | 12,681 | 6,834 | 6,678 | | 16,555 | | 18,605 | | 2,994 | |
| Curated micrographs (no.) | 5,820 | 4,659 | 3,657 | 5,983 | 10,220 | 6,700 | 2,383 | | 12,276 | | 18,201 | | 1,700 | |
| Initial particle images (no.) | 4,310,058 | 2,758,880 | 1,482,197 | 3,148,838 | 11,819,404 | 2,696,671 | 973,626 | | 4,184,713 | | 2,579,114 | | 502,895 | |
| Final particle images (no.) | 659,034 | 393,114 | 646,397 | 312,599 | 494,697 | 238,109 | 96,879 (core); 369,510 (guest) | | 1,320,184 (core); 882,884 (guest) | | 482,149 (core); 910,461 (guest) | | 54,897 (core); 158,181 (guest) | |
| Map resolution of core (Å) (FSC = 0.143) | 3.56 | 3.51 | - | - | - | 2.72 | 3.41 | | 2.71 | | 2.52 | | 3.35 | |
| Map resolution of guest (Å) (FSC = 0.143) | - | - | 3.67 | 2.87 | 3.01 | - | 3.49 | | 2.92 | | 2.87 | | 3.92 | |
| **Refinement$^‡$** | | | | | | | Core | Guest | Core | Guest | Core | Guest | Core | Guest |
| Model resolution (Å) (FSC = 0.143) | 3.4 | 3.4 | 3.6 | 2.9 | 3.0 | 2.7 | - | - | 2.7 | 2.9 | 2.5 | 2.9 | - | - |
| Map sharpening B factor (Å$^2$) | 117 | 100 | 85 | 41 | 53 | 52 | 70 | 97 | 23 | 49 | 56 | 53 | 58 | 92 |
| CC$_{mask}$ | 0.83 | 0.85 | 0.76 | 0.63 | 0.34 | 0.79 | - | - | 0.78 | 0.63 | 0.82 | 0.65 | - | - |
| No. atoms | | | | | | | | | | | | | | |
| RNA | 3996 | 3302 | 3493 | 1451 | 1756 | 8256 | - | - | 3872 | 1427 | 7828 | 1330 | - | - |
| Ligand | 0 | 0 | 0 | 0 | 29 | 0 | - | - | 0 | 24 | 0 | 12 | - | - |
| Ions/water | 42 | 56 | 30 | 12 | 0 | 352 | - | - | 392 | 20 | 429 | 15 | - | - |
| Mean B-factors (Å$^2$) | | | | | | | | | | | | | | |
| RNA | 165.4 | 197.6 | 140.0 | 40.6 | 37.6 | 96.2 | - | - | 28.2 | 38.7 | 77.7 | 52.7 | - | - |
| Ligand | - | - | - | - | 45.9 | - | - | - | - | 31.0 | - | 39.2 | - | - |
| Ions/water | 141.0 | 187.1 | 90.88 | 33.1 | - | 70.2 | - | - | 16.7 | 40.0 | 50.11 | 38.9 | - | - |
| r.m.s. deviations | | | | | | | | | | | | | | |
| Bond lengths (Å) | 0.007 | 0.008 | 0.006 | 0.006 | 0.007 | 0.006 | - | - | 0.007 | 0.008 | 0.006 | 0.009 | - | - |
| Bond angles (°) | 0.646 | 0.805 | 0.75 | 1.012 | 1.252 | 0.564 | - | - | 0.651 | 1.172 | 0.569 | 1.457 | - | - |

$^†$ Symmetry imposed only for core refinement.

$^‡$ Final particles for guest are C1 symmetry, and final core particles have symmetry imposed.

$^†$ Symmetry imposed only for core refinement.

**Extended Data Table 2 | Summary of cryoEM data symmetries, particle numbers and resolutions**

| Dataset | Symmetry | Particles | Resolution, Å |
|---|---|---|---|
| RSV1 | C2 | 659034 | 3.56 |
| RSV1 | C1 | 659034 | 3.79 |
| RSV2 (2OP) | C2 | 393114 | 3.51 |
| RSV2 (2OP) | C1 | 393114 | 4.03 |
| 2OP-Mango (tilted) | D2 | 96879 | 3.41 |
| 2OP-Mango (tilted) | C2 | 96879 | 3.68 |
| 2OP-Mango (tilted) | C1 | 96879 | 3.89 |
| 2OP-Quinine | C2 | 1320184 | 2.71 |
| 2OP-Quinine | C1 | 1320184 | 2.80 |
| 2OP-8oxoG | D2 | 482149 | 2.52 |
| 2OP-8oxoG | C2 | 482149 | 2.62 |
| 2OP-8oxoG | C1 | 482149 | 2.70 |
| 2OP-8oxoG (Glacios) | D2 | 54897 | 3.35 |
| 2OP-8oxoG (Glacios) | C2 | 54897 | 3.42 |
| 2OP-8oxoG (Glacios) | C1 | 54897 | 3.60 |

# Reporting Summary

## Statistics

For all statistical analyses, confirm that the following items are present in the figure legend, table legend, main text, or Methods section.

| n/a | Confirmed | |
|---|---|---|
| ☒ | ☐ | The exact sample size (*n*) for each experimental group/condition, given as a discrete number and unit of measurement |
| ☒ | ☐ | A statement on whether measurements were taken from distinct samples or whether the same sample was measured repeatedly |
| ☒ | ☐ | The statistical test(s) used AND whether they are one- or two-sided *Only common tests should be described solely by name; describe more complex techniques in the Methods section.* |
| ☒ | ☐ | A description of all covariates tested |
| ☒ | ☐ | A description of any assumptions or corrections, such as tests of normality and adjustment for multiple comparisons |
| ☒ | ☐ | A full description of the statistical parameters including central tendency (e.g. means) or other basic estimates (e.g. regression coefficient) AND variation (e.g. standard deviation) or associated estimates of uncertainty (e.g. confidence intervals) |
| ☒ | ☐ | For null hypothesis testing, the test statistic (e.g. *F*, *t*, *r*) with confidence intervals, effect sizes, degrees of freedom and *P* value noted *Give P values as exact values whenever suitable.* |
| ☒ | ☐ | For Bayesian analysis, information on the choice of priors and Markov chain Monte Carlo settings |
| ☒ | ☐ | For hierarchical and complex designs, identification of the appropriate level for tests and full reporting of outcomes |
| ☒ | ☐ | Estimates of effect sizes (e.g. Cohen's *d*, Pearson's *r*), indicating how they were calculated |

*Our web collection on statistics for biologists contains articles on many of the points above.*

## Software and code

Policy information about availability of computer code

| Data collection | CryoEM data was collected with standard software packages, such as SerialEM. |
|---|---|
| Data analysis | CryoEM data was analyzed with standard software packages, such as RELION and cryoSPARC. |

For manuscripts utilizing custom algorithms or software that are central to the research but not yet described in published literature, software must be made available to editors and reviewers. We strongly encourage code deposition in a community repository (e.g. GitHub). See the Nature Portfolio guidelines for submitting code & software for further information.

## Data

Policy information about availability of data

All manuscripts must include a data availability statement. This statement should provide the following information, where applicable:
- Accession codes, unique identifiers, or web links for publicly available datasets
- A description of any restrictions on data availability
- For clinical datasets or third party data, please ensure that the statement adheres to our policy

Maps were deposited to the Electron Microscopy Data Bank following recent guidelines24, and atomic coordinates were deposited in the Protein Data Bank (accession codes in Supplementary Extended Data Table 3). Movies for each dataset were deposited in EMPIAR. These include micrographs for RSV1 (EMPIAR-12578/data/Op1), RSV2 (EMPIAR-12578/data/Op2), 2OP-tRNAAsp (EMPIAR-12578/data/tRNA), 2OP-Mango (no ligand) (EMPIAR-12578/data/Mango/NoLigand), 2OP-Mango (with ligand) (EMPIAR-12578/data/Mango/Ligand), 2OP-U1A (EMPIAR-12578/data/U1A), 2OP-Mango (no ligand) collected at a 45° stage tilt

(EMPIAR-12578/data/Mango/Tilted), 2OP-Quinine (EMPIAR-12578/data/Quinine), 2OP-8oxoG (EMPIAR-12578/data/8oxoG), and 2OP-8oxoG-Glacios (EMPIAR-12578/data/8oxoG-Glacios).

## Research involving human participants, their data, or biological material

Policy information about studies with human participants or human data. See also policy information about sex, gender (identity/presentation), and sexual orientation and race, ethnicity and racism.

| | |
|---|---|
| Reporting on sex and gender | *Use the terms sex (biological attribute) and gender (shaped by social and cultural circumstances) carefully in order to avoid confusing both terms. Indicate if findings apply to only one sex or gender; describe whether sex and gender were considered in study design; whether sex and/or gender was determined based on self-reporting or assigned and methods used. Provide in the source data disaggregated sex and gender data, where this information has been collected, and if consent has been obtained for sharing of individual-level data; provide overall numbers in this Reporting Summary. Please state if this information has not been collected. Report sex- and gender-based analyses where performed, justify reasons for lack of sex- and gender-based analysis.* |
| Reporting on race, ethnicity, or other socially relevant groupings | *Please specify the socially constructed or socially relevant categorization variable(s) used in your manuscript and explain why they were used. Please note that such variables should not be used as proxies for other socially constructed/relevant variables (for example, race or ethnicity should not be used as a proxy for socioeconomic status). Provide clear definitions of the relevant terms used, how they were provided (by the participants/respondents, the researchers, or third parties), and the method(s) used to classify people into the different categories (e.g. self-report, census or administrative data, social media data, etc.) Please provide details about how you controlled for confounding variables in your analyses.* |
| Population characteristics | *Describe the covariate-relevant population characteristics of the human research participants (e.g. age, genotypic information, past and current diagnosis and treatment categories). If you filled out the behavioural & social sciences study design questions and have nothing to add here, write "See above."* |
| Recruitment | *Describe how participants were recruited. Outline any potential self-selection bias or other biases that may be present and how these are likely to impact results.* |
| Ethics oversight | *Identify the organization(s) that approved the study protocol.* |

Note that full information on the approval of the study protocol must also be provided in the manuscript.

# Field-specific reporting

Please select the one below that is the best fit for your research. If you are not sure, read the appropriate sections before making your selection.

☒ Life sciences ☐ Behavioural & social sciences ☐ Ecological, evolutionary & environmental sciences

For a reference copy of the document with all sections, see nature.com/documents/nr-reporting-summary-flat.pdf

# Life sciences study design

All studies must disclose on these points even when the disclosure is negative.

| | |
|---|---|
| Sample size | Sample sizes in this study could be considered the number of micrographs collected (Extended Data Table 2). These numbers were determined after analysis of the data to see if sufficient map resolution was obtained. The overall number collected was largely determined by the allotment of microscope time (i.e., 4 days). |
| Data exclusions | Icy micrographs and bad particles were excluded as is customary for this type of analysis. |
| Replication | Individual datasets were analyzed as is and not replicated. |
| Randomization | Randomization is not relevant for this study. |
| Blinding | Blinding is not relevant for this study. |

# Reporting for specific materials, systems and methods

We require information from authors about some types of materials, experimental systems and methods used in many studies. Here, indicate whether each material, system or method listed is relevant to your study. If you are not sure if a list item applies to your research, read the appropriate section before selecting a response.

## Materials & experimental systems

| n/a | Involved in the study |
|---|---|
| ☒ ☐ | Antibodies |
| ☒ ☐ | Eukaryotic cell lines |
| ☒ ☐ | Palaeontology and archaeology |
| ☒ ☐ | Animals and other organisms |
| ☒ ☐ | Clinical data |
| ☒ ☐ | Dual use research of concern |
| ☒ ☐ | Plants |

## Methods

| n/a | Involved in the study |
|---|---|
| ☒ ☐ | ChIP-seq |
| ☒ ☐ | Flow cytometry |
| ☒ ☐ | MRI-based neuroimaging |

## Plants

| Seed stocks | Not applicable |
|---|---|
| Novel plant genotypes | Not applicable |
| Authentication | Not applicable |

