## [Peer Review File · Nature Methods]

Scaffolds with optimized quaternary symmetry for de novo cryoEM structure determination of small RNAs

Corresponding Author: Dr Christopher Jones

Version 0:

Decision Letter:

27th May 2025

Dear Dr. Jones,

Your Article, "Scaffolds with optimized quaternary symmetry for de novo cryoEM structure determination of small RNAs", has now been seen by 3 reviewers. As you will see from their comments below, although the reviewers find your work of considerable potential interest, they have raised a number of concerns. We are interested in the possibility of publishing your paper in Nature Methods, but would like to consider your response to these concerns before we reach a final decision on publication.

We therefore invite you to revise your manuscript to address these concerns. It would be important to discuss the limitations of the work and to place the methodological advance more carefully in the context of other methods in this space. We recommend careful revisions to ensure all reviewer concerns are alleviated.

Link Redacted

We hope to receive your revised paper within 6-8 weeks. If you cannot send it within this time, please let us know. In this event, we will still be happy to reconsider your paper at a later date so long as nothing similar has been accepted for publication at Nature Methods or published elsewhere.

OPEN SCIENCE REQUIREMENTS

REPORTING SUMMARY AND EDITORIAL POLICY CHECKLISTS

IMAGE INTEGRITY

EXTENDED DATA FIGURES

DATA AVAILABILITY

All novel DNA and RNA sequencing data, protein sequences, genetic polymorphisms, linked genotype and phenotype data, gene expression data, macromolecular structures, and proteomics data must be deposited in a publicly accessible database, and accession codes and associated hyperlinks must be provided in the "Data Availability" section.

Please include a "Data availability" subsection in the Online Methods. This section should inform readers about the availability of the data used to support the conclusions of your study, including accession codes to public repositories, references to source data that may be published alongside the paper, unique identifiers such as URLs to data repository entries, or data set DOIs, and any other statement about data availability. At a minimum, you should include the following statement: "The data that support the findings of this study are available from the corresponding author upon request", describing which data is available upon request and mentioning any restrictions on availability. If DOIs are provided, please include these in the Reference list

(authors, title, publisher (repository name), identifier, year). For more guidance on how to write this section please see: <http://www.nature.com/authors/policies/data/data-availability-statements-data-citations.pdf>

MATERIALS AVAILABILITY

SUPPLEMENTARY PROTOCOL

To help facilitate reproducibility and uptake of your method, we ask you to prepare a step-by-step Supplementary Protocol for the method described in this paper. We [encourage authors to share their step-by-step experimental protocols](https://www.nature.com/nature-research/editorial-policies/reporting-standards#protocols) on a protocol sharing platform of their choice and report the protocol DOI in the reference list. Nature Portfolio's protocols.io is a free-to-use and open resource for protocols; protocols deposited onto protocols.io are citable and can be linked from the published article. More details can found at [protocols.io](https://www.protocols.io/help/publish-articles).

ORCID

Nature Methods is committed to improving transparency in authorship. As part of our efforts in this direction, we are now requesting that all authors identified as 'corresponding author' on published papers create and link their Open Researcher and Contributor Identifier (ORCID) with their account on the Manuscript Tracking System (MTS), prior to acceptance. This applies to primary research papers only. ORCID helps the scientific community achieve unambiguous attribution of all scholarly contributions. You can create and link your ORCID from the home page of the MTS by clicking on 'Modify my Springer Nature account'. For more information please visit [please visit](http://www.springernature.com/orcid) www.springernature.com/orcid.

Sincerely,
Arunima

Arunima Singh, Ph.D.
Senior Editor
Nature Methods

Reviewers' Comments:

Reviewer #1 (Remarks to the Author):

This manuscript presents a creative method to facilitate RNA structure determination using cryoEM, which is of great interest given the interest and importance of RNA structure and the paucity of RNA structural information compared to proteins. Crystallography has served RNA structural biology well, but in recent years cryoEM has presented a powerful alternative. However, many challenges remain before RNA structures can be determined routinely to high resolution using cryoEM. Many biological important RNAs are relatively small and such targets remain a challenge for cryoEM. Here, the authors present a new method that makes use of a multimeric scaffold of known structure on which the target RNA is appended for structure determination. Previously presented scaffold-based methods have been successfully demonstrated, but those used larger monomeric RNAs as the scaffold. The use of a multimeric and perhaps smaller RNA may have some advantages or at least can complement other scaffold-based methods. Overall, the work presented here is of high quality creative, and interesting. I think there is definite value in having several scaffold-based methods available to allow continued development, optimization, and to match each target to the appropriate scaffold. The field needs more tools, and more ideas on which to build additional tools. With that, I do have some criticisms that should be addressed before publication.

1. The real value of the symmetry is not clear. The authors make some general statements about the value of symmetry (which I agree with), and they state in a few places that the symmetric multimer gave better maps than the monomer parent molecule, but these are somewhat ambiguous. Was this primarily because of the increased size, less ambiguous particle picking, better alignment, or actual application of symmetry in map refinement? The authors should show comparisons of maps with and without symmetry applied, and more detailed workflow figures, which is standard now for all structures solved by cryoEM. My prediction is that the symmetry helped the quality of the maps associated with the scaffold more than the appended target, but I stand to be corrected if those data and comparison maps are provided. As symmetry is potentially what sets this scaffold

method apart from others and given that this is a “methods” paper, much more rigorous analysis of its value in each case needs to be presented.

2. Related to the above, flow charts that illustrate the workflow, improvements to the maps, number of particles at each step, etc. need to be included.
3. There are no direct comparisons of the target structures with and without the scaffold. While this may not be possible for the smaller RNAs that are very hard to pick and align, I suspect tRNA could be done without a scaffold. How much does the scaffold help in this case?
4. Were the maps good enough to build the structures ab initio? The two riboswitch maps appear to be quite good in their centers, but if previously solved riboswitch structures were not available, and the location of the ligand was not known, were these maps adequate to determine the locations of all bases and the ligand? Note that if the answer is “no,” this does not preclude publication or the value of the method, but it does put a (current) limit on its application that should be acknowledged.
5. The use of a G-G mismatch in the connecting stem of two of the targets is puzzling to me. I would think that if the maps are good, there is no need for a fiduciary (one can just count base pairs from the scaffold) and such a mismatch is likely to increase local flexibility, which decreases map resolution. Did the authors try RNAs without this marker? Were they better? Worse?
6. Minor: The legend for the “colored by resolution maps” are from “best to worst” which is not very informative – they should have actual numbers.

Note that all my comments are within this critique, there are no confidential comments to the editors.

Reviewer #1 (Remarks on figshare data availability):

None

Reviewer #2 (Remarks to the Author):

The manuscript of Jones and Ferre-D’Amare describes an approach to overcome sample size limitations in the study of the structure of small RNA molecules by cryoEM. The method uses an RNA scaffold to attach a small RNA molecule (guest) as an extension of a stem in the scaffold. The scaffold chosen, the Rous Sarcoma Virus Frameshifting Stimulatory Element (RSV FSE), has the advantage that it is medium size and it dimerizes readily and sometimes it even tetramerizes to have either C2 or D2 symmetry. The idea of using RNA scaffolds to study smaller RNAs is not new and has been done before, although using different scaffolds, and with mixed results. The suggested advantage of the RSV FSE scaffold is the inherent symmetry, which helps in structure determination, its size, and the ease to introduce the desired elements. One major limitation is that only RNA molecules that have the 5’ and 3’ ends close in space, for example by forming a stem, are suitable for this approach. Whereas there are many RNA fragments that do have these characteristics, it is by no means a universal feature of RNA molecules. The authors tested their approach with different targets and the results are mixed. In many cases, the resolution achieved for the targets is in the medium resolution range (3.5 Å or worse). This is due to the intrinsic mobility and flexibility of RNA molecules, which has always been a limitation in RNA structural studies. Attaching additional elements to the end of the stem seems to result in a region that is moving enough to degrade the local resolution. The core of the scaffold may be rigid, but that does not help when the elements sticking out are moving. In many of the examples, there are poorly resolved areas and in some cases it was not possible to trace the chain unambiguously. The best results in terms of resolution come from the smallest elements studied. These observations suggest that the RSV FSE scaffold is an interesting and potentially powerful tool, but with limited applicability and confined to rigid and relatively small RNA molecules.

There is a real need for new methods to study the structure of RNA molecules and this approach could be very useful in this regard, but the authors should describe and discuss the limitations of their approach and temper their claims of resolution achieved. The Abstract claims that “routinely enable de novo structure solution of covalently attached RNA guest to beyond 3 Å overall resolution.” This statement does not reflect their results. In most cases the guest structures were not close to 3 Å and higher resolution was only achieved in limited regions. Overall, this is an interesting approach and one that could be very useful, but there are limitations that need to be discussed and a more candid description of the results is needed. In addition, some more specific points are listed below.

1. It would help to have a table that lists the RNA guest molecules, the overall resolution attained and the resolution for the guests, not just the overall resolution. The important piece of information that needs to be highlighted is the resolution of the guests and completeness of the models for the guest molecules, not the overall numbers. The information is buried in the Extended Data. Instead, it should be a more prominent single Table.
2. The Methods section is long and detailed. Some of the details are not needed and the section could be more concise and focus on the aspects relevant to the scaffolding approach.
3. In the figures, such as Figure 4, it is not always clear where the guest molecule is found in the sequence and the structure. It should be made clearer.
4. In the Extended Data Figures, it is necessary to add resolution to the color gradients. Best and Worst are not quantitative enough and do not allow proper assessment and comparison. Resolution numbers should be added to allow readers to assess the quality of the structures.
5. In several of the FSC curves shown there is a dip in the FSC. This is probably due to selecting a mask that is too tight or small. The authors need to correct or discuss this. It may reflect problems with data processing.
6. As mentioned above, the Extended Figures should label the region corresponding to the guest molecules. It is not always obvious where the guests are as the orientations shown are not consistent.
7. Extended Figure 2 mixes information on the Mango structure and the tRNA structure. All the Mango information belongs to Extended Figure 3 to separate clearly the two cases.

Reviewer #2 (Remarks on figshare data availability):

The data shows SEC traces, which are also shown in the paper. They appear to agree.

Reviewer #3 (Remarks to the Author):

This article by Jones and Ferre d'Amaré proposes a novel multimeric scaffolded method that is based on the structure of the RSV frameshifting element to routinely enable the de novo structure solution of covalently appended small RNAs by cryo-EM at resolution of 3 Å. This approach is particularly attractive as the scaffold used in this study is quite small in size but because of its ability of forming either dimers or tetramers, it still allows good atomic resolutions of various guest RNAs fused to it. The method proposed is quite novel and has the potential to contribute to solve the structures of many other small guest RNAs. This work definitely increases the tool box of RNA scaffolds that could be used for solving the structures of small RNAs that lack density for being able to be studied by cryo-EM. The experimental work and data analysis have been performed with much care and overall, the article is well written with many helpful figures. Because of its novelty, this article would certainly be suitable for publication in Nature methods. Nevertheless, it also raises few outstanding questions mentioned below that would be worth exploring further in order to increase further the potential of the proposed approach to be used by other cryo-EM laboratories.

Major comments

The guest RNA is inserted into a region (P3) of the RSV scaffold that is somewhat oriented away from the RSV core. However, the nature of the guest RNA seems to contribute somehow to the supramolecular assembly into either C2 or D2 symmetric particles. This is quite intriguing and it would be helpful if some rational could be proposed, especially from a methodological, rational engineering point of view. Is there any possible control over the formation of D2 versus C2 symmetric particles? What are the differences in conditions, processes that might favor the formation of C2 symmetric particles versus D2 symmetric particles or vice versa?

The quinine aptamer and oxoG riboswitch are structurally related. Why does the 2OP quinine construct form mostly dimers while the 2OP-8oxoG assembles in particles with tetrameric D2 symmetry? Is this due to the 2bps difference in length of the connector stem P3 that connects the guest RNA to the scaffold?

While the atomic resolutions of the quinine aptamer and oxoG riboswitch are outstanding, the resolution of the i-mango is not too great and it is somewhat surprising that the UA1 RNA protein complex is not able to be resolved with the proposed method. As the atomic structure of the i-mango aptamer and the UA1 RNP complex have been previously solved, what could be the factors behind the suboptimal resolution of these structures versus the two last ones (quinine aptamer and oxoG riboswitch)? Could this be due to variations in the way the folding protocols were performed, methodology, salt concentrations, presence of different ligands, or RNA engineering with P3 connectors being not optimal in length?

While the answers to these questions might still be unclear as noted by the authors at the end of the discussion, it would be good to provide some clues in order to increase the likelihood of success of the proposed methodology once used by other laboratories.

Minor comments

Lane 116 p.7: it is 2OP rather than OP2.

Reviewer #3 (Remarks on figshare data availability):

The work has been done with great expertise and is clearly supporting the claims of the article.

Version 1:

Decision Letter:

Our ref: NMETH-A60215A

9th Sep 2025

Dear Chris,

Thank you for submitting your revised manuscript "Scaffolds with optimized quaternary symmetry for de novo cryoEM structure determination of small RNAs" (NMETH-A60215A). It has now been seen by the original referees and their comments are below. The reviewers find that the paper has improved in revision, and therefore we'll be happy in principle to publish it in Nature Methods, pending minor revisions to satisfy the referees' final requests and to comply with our editorial and formatting guidelines. I recommend addressing these final requests from the authors and also toning down the claims made in the paper regarding the method.

TRANSPARENT PEER REVIEW

ORCID

Sincerely,
Arunima

Arunima Singh, Ph.D.
Senior Editor
Nature Methods

Reviewer #1 (Remarks to the Author):

Thank you for the opportunity to review the revised version of this manuscript. In my previous review, I included a summary of the results and therefore will not articulate them here. My primary concerns in my review of the first submission were that the value of the symmetry was not clear or adequately discussed, some additional detail regarding the cryoEM workflow, the need for the fiducial, and direct comparison of maps with and without the scaffold attached.

Overall, my concerns have been mostly adequately addressed and this is suitable for publication. I remain a bit skeptical that the symmetry is really adding much - I think it is almost certainly that the particles are larger and easier to align, but that is not a concern that precludes publication. I also think long-term application and benefit of the method will require much more optimization and examination of different RNAs beyond riboswitches, but this will likely be something that the field will work out in the coming years.

The one thing I remain a bit unhappy about is the lack of a map color coded by resolution. While I understand that a recent publication recommended using "highest to lowest" maps, such maps still give a range for the resolution using current statistical methods and are useful if even from a quantitative visualization and comparison. It is hard to justify referring in the text to the resolution of the final maps that the scaffold method obtained, and claim improvements of 0.1 Angstroms, and then not include a map because the numbers are not "quantitatively reliable." However, as this was a minor concern in my first review, I won't hold up publication over this.

Reviewer #2 (Remarks to the Author):

The revised manuscript has addressed many concerns appropriately, but I still have some remaining concerns and suggestions. In general, the manuscript still exaggerates the resolution attained by the method. This is a serious concern as a read of the Abstract gives the false impression that it routinely attains resolutions over 3A, which is not the case. See the comments below.

1. Table 1 Summarizes the results, including the resolution. The table is incomplete as it does not include the results for tRNA or U1A, both cases where the resolution was poorer. In addition, the information that is most relevant, the resolution of the guest, is not included. The table should include the overall estimated resolution for host plus guest and in a separate column the resolution for the guest.

2. Of the structures that are the test cases, U1A, tRNA, Mango, Quinine riboswitch, and 8oxoG riboswitch, only two of them achieve resolutions better than 3.0 A for the guest (2.9 A for both 2OP-Quinine and 2OP-8oxoG). The other 3 structures range from 3.7 – 3.5 A and unknown (U1A). For this reason, the statement in the Abstract "... enable de novo structure solution of covalently attached guest to beyond 3A overall resolution" is misleading. The sentence must reflect the results. In some cases the overall resolution of the guest extended just beyond 3 A, but in most cases a more modest resolution was attained.

3. The most important thing that researchers want to know when assessing a method for solving a new RNA structure is whether it is possible to trace the structure de novo. It seems that this was only true in very few cases. This information should also be included in Table 1. This is important as the manuscript includes statements like “obtaining reconstructions at ~3.0 Å resolution for both ...” (Mango structures) followed by “Neither map was of sufficient quality to fully trace the intricate aptamer...”. Clearly the aptamer resolution was much worse than 3.0 Å, but an inexperienced reader may wrongly infer that the guest structure was at 3.0 Å and yet still untraceable.

4. The reason why there is a recommendation not to use numerical values for local resolution estimate maps is that the methods to calculate local resolution are unreliable, not because this is unwanted or irrelevant information (“The community consensus appears to be that estimated local-resolution values are not quantitatively reliable. Thus, the global resolution estimate could be supplemented by a coloured local-resolution map accompanied by a colour legend labelled not with specific Ångström values, but rather ‘better’ (blue) to ‘worse’ (red) resolution. Finally, it was recognized that, for cases where an atomic model is available, a visual depiction of local resolution across the amino-acid or nucleotide sequence would be a valuable addition to future validation reports, once robust algorithms that produce such a mapping become available and are widely accepted in the community.”) I suggest that at the very least the color legend should show the resolution value estimated for the structure. For example, in Extended Data Figure 3, the color ramps should have best and worse plus the actual estimated resolution values, in this case 2.9 Å and 3.0 Å marked in the color ramps. In this way the readers can have a reference for what color corresponds to the estimated resolution.

5. I still think that in every figure the guest molecule should be clearly marked. Referring to Extended Data Figure 3 again, the Mango portion of the structures should be boxed to make it very clear where this portion is. This is very important to assess the resolution of the guest portion of the molecule in relation to the host.

6. There are some minor errors that could be removed by careful editing. For example, in the caption of Extended Data Figure 5, it says “a map threshold of 10.” I think they mean a map threshold of 10 Sigma.

Reviewer #3 (Remarks to the Author):

All the comments of the referee have been addressed.

Reviewer #3 (Remarks on figshare data availability):

None

Version 2:

Decision Letter:

26th Jan 2026

Dear Chris,

I am pleased to inform you that your Article, "Scaffolds with optimized quaternary symmetry for de novo cryoEM structure determination of small RNAs", has now been accepted for publication in Nature Methods. The received and accepted dates will be March 14, 2025 and January 26, 2026. This note is intended to let you know what to expect from us over the next month or so, and to let you know where to address any further questions.

Over the next few weeks, your paper will be copyedited to ensure that it conforms to Nature Methods style. Once your paper is typeset, you will receive an email with a link to choose the appropriate publishing options for your paper and our Author Services team will be in touch regarding any additional information that may be required. It is extremely important that you let us know now whether you will be difficult to contact over the next month. If this is the case, we ask that you send us the contact information (email, phone and fax) of someone who will be able to check the proofs and deal with any last-minute problems.

Authors may need to take specific actions to achieve compliance with funder and institutional open access mandates.

If your research is supported by a funder that requires immediate open access (e.g. according to [a href="https://www.springernature.com/gp/open-science/plan-s-compliance"> Plan S principles](https://www.springernature.com/gp/open-science/plan-s-compliance) or the [a href="https://www.springernature.com/gp/open-science/us-federal-agency-compliance"> NIH public access policy](https://www.springernature.com/gp/open-science/us-federal-agency-compliance)) then you should select the gold OA route, and we will direct you to the compliant route where possible. Because authors warrant under our subscription licensing terms that they haven't committed to licensing any version of their article under a licence inconsistent with the terms of our agreement – including the applicable embargo period – publication under the subscription model isn't suitable for authors whose funders require no embargo.

If you have posted a preprint on any preprint server, please ensure that the preprint details are updated with a publication

reference, including the DOI and a URL to the published version of the article on the journal website.

If you are active on Twitter/X or Bluesky, please e-mail me your and your coauthors' handles so that we may tag you when the paper is published.

Best regards,
Arunima

Arunima Singh, Ph.D.
Senior Editor
Nature Methods

** Visit the Springer Nature Editorial and Publishing website at http://editorial-jobs.springernature.com?utm_source=ejP_NMeth_email&utm_medium=ejP_NMeth_email&utm_campaign=ejp_Nmeth for more information about our career opportunities. If you have any questions please click [here](mailto:editorial.publishing.jobs@springernature.com).**

Response to Reviewers

The reviewers' comments are black, and our point-by-point response are red (textual changes to the MS file are also in red).

Reviewers' Comments:

Reviewer #1 (Remarks to the Author):

This manuscript presents a creative method to facilitate RNA structure determination using cryoEM, which is of great interest given the interest and importance of RNA structure and the paucity of RNA structural information compared to proteins. Crystallography has served RNA structural biology well, but in recent years cryoEM has presented a powerful alternative. However, many challenges remain before RNA structures can be determined routinely to high resolution using cryoEM. Many biological important RNAs are relatively small and such targets remain a challenge for cryoEM.

Here, the authors present a new method that makes use of a multimeric scaffold of known structure on which the target RNA is appended for structure determination. Previously presented scaffold-based methods have been successfully demonstrated, but those used larger monomeric RNAs as the scaffold. The use of a multimeric and perhaps smaller RNA may have some advantages or at least can complement other scaffold-based methods.

Overall, the work presented here is of high quality creative, and interesting. I think there is definite value in having several scaffold-based methods available to allow continued development, optimization, and to match each target to the appropriate scaffold. The field needs more tools, and more ideas on which to build additional tools.

With that, I do have some criticisms that should be addressed before publication.

1. The real value of the symmetry is not clear. The authors make some general statements about the value of symmetry (which I agree with), and they state in a few places that the symmetric multimer gave better maps than the monomer parent molecule, but these are somewhat ambiguous. Was this primarily because of the increased size, less ambiguous particle picking, better alignment, or actual application of symmetry in map refinement?

The authors should show comparisons of maps with and without symmetry applied, and more detailed workflow figures, which is standard now for all structures solved by cryoEM. My prediction is that the symmetry helped the quality of the maps associated with the scaffold more than the appended target, but I stand to be corrected if those data and comparison maps are provided. As symmetry is potentially what sets this scaffold method apart from others and given that this is a “methods” paper, much more rigorous analysis of its value in each case needs to be presented.

We thank the reviewer for the insightful comments. It is challenging to pinpoint the contributions of each of these factors (size, picking, alignment, symmetry) and disentangle them. However, studies on small RNAs suggest that size in part limits their resolution, and in our experience, small C1 RNA particles are very difficult to align (see our SL5 paper just published in RNA while this MS was under review, which we have now cited). We have also included a citation to another manuscript examining tRNA modifications via cryoEM published while our paper was

under review (addressed below in point 3). The comparison between that study on free tRNAs and ours at least suggests that scaffolding is sufficient to increase resolution, at minimum, for tRNAs.

As the reviewer suggests, C2 or D2 symmetry is applied for all final core local refinements (final steps) as the maps still possess internal symmetry. In addition, they have been applied for maps prior to local refinement during the analysis process. However, for guest RNA local refinements after symmetry expansion, symmetry is no longer applied as these maps are either no longer symmetric due to particle subtraction or inappropriate due to particle duplication during symmetry expansion. During local refinement of guests, the “use gaussian priors” option in cryoSPARC is turned on, so having better alignments of scaffolds prior to these steps should improve alignments during local refinement.

To partly address these questions, we have included C1 (i.e., no symmetry) refinements for the core maps now, which are typically lower resolution. These are summarized in a new Table 1. As expected, symmetry provides a resolution improvement that is most noticeable for lower resolution datasets. For higher resolution datasets, each step in symmetry is an ~ 0.1 Å improvement.

A new paragraph in the discussion section now reads:

"Use of scaffolds for cryoEM structure determination can impact, in principle, all facets of the methodology, including sample stability and folding, particle picking, alignment, averaging, etc. To what extent the increased molecular mass of the sample, its improved stability and homogeneity, and the symmetry of the particle result in improved final reconstructions and atomic models is difficult to deconvolute. Recent reconstructions of free tRNA³⁴ are among the smallest RNA molecules investigated to date by single-particle cryoEM and were limited to 4.5-5 Å resolution. In comparison, our reconstruction, using the 2-fold symmetric scaffold, has a resolution of ~ 3.7 Å. In our approach, C2 or D2 symmetry is applied for all final core local refinements as the maps still possess internal symmetry. In addition, symmetry has been applied for maps prior to local refinement during the analysis process. However, for guest RNA local refinements after symmetry expansion, symmetry is no longer applied as these maps are either no longer symmetric due to particle subtraction or inappropriate due to particle duplication during symmetry expansion. It is thus possible that symmetry is particularly useful for lower resolution datasets. Comparison of core refinements with and without (C1 symmetry) suggests that for higher-resolution datasets, each doubling of symmetry improves resolution by ~ 0.1 Å (Table 1)."

2. Related to the above, flow charts that illustrate the workflow, improvements to the maps, number of particles at each step, etc. need to be included.

We have included flowcharts now for the 2OP-Quinine and 2OP-8oxoG datasets (new Extended Data Figures 7 and 9). The overall process was similar between the datasets.

3. There are no direct comparisons of the target structures with and without the scaffold. While this may not be possible for the smaller RNAs that are very hard to pick and align, I suspect tRNA could be done without a scaffold. How much does the scaffold help in this case?

A paper describing free tRNA structures using cryoEM was published while our MS was under review, and they obtained 4.5-5 Å resolution maps (i.e., worse than those here). As this is the largest RNA attached to the scaffold, it is the most likely to succeed in free form. We have revised our manuscript to cite

Ref 34: Biela, A.D. et al. Determining the effects of pseudouridine incorporation on human tRNAs. *EMBO J* **44**, 3553-3585 (2025).

This is part of the new Discussion paragraph mentioned in response to Point 1, above.

4. Were the maps good enough to build the structures ab initio? The two riboswitch maps appear to be quite good in their centers, but if previously solved riboswitch structures were not available, and the location of the ligand was not known, were these maps adequate to determine the locations of all bases and the ligand? Note that if the answer is “no,” this does not preclude publication or the value of the method, but it does put a (current) limit on its application that should be acknowledged.

The quinine aptamer and 8-oxoguanine aptamers were good enough to be traced *de novo* in Coot, and the ligand binding sites were apparent even before tracing by examining the maps. As the ligand binding site regions are very good in each map, as shown in the figures, the locations and poses of each ligand were built *de novo*. The Mango and tRNA structures would have been more challenging as they are lower resolution, and these builds were not attempted *de novo* (also as structures are available for each). The worst regions of each map are of course more challenging to build unambiguously, and this is the case for all maps. The local resolution estimates are provided to indicate confidence along these lines.

5. The use of a G-G mismatch in the connecting stem of two of the targets is puzzling to me. I would think that if the maps are good, there is no need for a fiducial marker (one can just count base pairs from the scaffold) and such a mismatch is likely to increase local flexibility, which decreases map resolution. Did the authors try RNAs without this marker? Were they better? Worse?

This work was initiated over two years ago, at which time the quality of RNA maps were considerably worse than they are now, even in our own hands (~3.5 Å was the best we could do). At that time, the fiducial marker was essential to convince ourselves of map quality. Resolution variation is likely due to many reasons, but this concern was greater before sub-3 Å maps were obtained. It is reasonable to hypothesize that removing these markers might improve rigidity, but even with them, the complex is sufficiently rigid enough to enable structure determination. We suspect application of 2OP scaffolds would not require these markers, but we have not systematically tested them without markers. We now include a clarifying statement in the “*Application of the optimized quaternary scaffold to tRNA^{ASP} and iMango-III A10U*” section, thus:

“We expected that the fiducial marker would be particularly important when the resolution of the final map was not sufficiently high. As the mismatch may be destabilizing, it could be omitted if higher-resolution RNA reconstructions become routine.”

6. Minor: The legend for the “colored by resolution maps” are from “best to worst” which is not very informative – they should have actual numbers.

This “best to worse” coloring is recommended by recent cryoEM guidelines cited in the MS (<https://pubmed.ncbi.nlm.nih.gov/38358351/>, recommendation #14), as the numbers are not quantitatively reliable. We have added the following to the first callout of local resolution maps:

“The “best to worse” resolution designations (Extended Data Fig. 1d,e and all other local resolution maps) follow recently published community recommendations for cryoEM reporting.²⁴”

And to the Extended Data Fig. 1 caption:

“The “best to worse” resolution designations follow recently published community recommendations for cryoEM reporting.”

Note that all my comments are within this critique, there are no confidential comments to the editors.

Reviewer #1 (Remarks on figshare data availability):

None

Reviewer #2 (Remarks to the Author):

The manuscript of Jones and Ferre-D’Amare describes an approach to overcome sample size limitations in the study of the structure of small RNA molecules by cryoEM. The method uses an RNA scaffold to attach a small RNA molecule (guest) as an extension of a stem in the scaffold. The scaffold chosen, the Rous Sarcoma Virus Frameshifting Stimulatory Element (RSV FSE), has the advantage that it is medium size and it dimerizes readily and sometimes it even tetramerizes to have either C2 or D2 symmetry. The idea of using RNA scaffolds to study smaller RNAs is not new and has been done before, although using different scaffolds, and with mixed results.

It has been done before with largely *poor* results. Only recently did one method produce sub-3 Å maps for previously determined RNA structures. This is the first case of applying the method to produce new structures of good quality for RNA. We now include a table in supplement comparing the nominal resolution limits of various approaches.

The suggested advantage of the RSV FSE scaffold is the inherent symmetry, which helps in structure determination, its size, and the ease to introduce the desired elements. One major limitation is that only RNA molecules that have the 5' and 3' ends close in space, for example by forming a stem, are suitable for this approach. Whereas there are many RNA fragments that do have these characteristics, it is by no means a universal feature of RNA molecules.

The 5' and 3' ends are adjacent to one another in the scaffold, meaning it is easy to permute this RNA for the study of pseudoknots, which would be attached at new 5' and 3' ends present in helix P3. We mention this in the discussion:

“For pseudoknots, one could envision using a permuted version of our RSV FSE-derived scaffold, joining the natural 5' and 3' ends (Fig. 1a), which are juxtaposed and stack on one another in dimers. Placing new 5' and 3' ends at P3 would thus allow for inclusion of a pseudoknot RNA at P3 or even "sticky" ends for adding guests *in trans*. While we have explored both options (data not shown), they require further optimization.”

The authors tested their approach with different targets and the results are mixed. In many cases, the resolution achieved for the targets is in the medium resolution range (3.5 Å or worse). This is due to the intrinsic mobility and flexibility of RNA molecules, which has always been a limitation in RNA structural studies. Attaching additional elements to the end of the stem seems to result in a region that is moving enough to degrade the local resolution. The core of the scaffold may be rigid, but that does not help when the elements sticking out are moving. In many of the examples, there are poorly resolved areas and in some cases it was not possible to trace the chain unambiguously. The best results in terms of resolution come from the smallest elements studied.

This is inaccurate, as the smallest guest RNAs attached—the U1A stemloop and Mango quadruplex—were more poorly resolved than the quinine aptamer and 8-oxoguanine aptamer, which were better resolved than larger tRNA^{Asp}. As the reviewer notes, solution flexibility plays an important role here in limiting resolution for some specimens. This does not simply reflect size, unfortunately, or crystal structure resolution (as Mango is by far the highest). This is addressed in the new Discussion paragraph (response to Point 1 of Referee 1, above).

These observations suggest that the RSV FSE scaffold is an interesting and potentially powerful tool, but with limited applicability and confined to rigid and relatively small RNA molecules. There is a real need for new methods to study the structure of RNA molecules and this approach could be very useful in this regard, but the authors should describe and discuss the limitations of their approach and temper their claims of resolution achieved. The Abstract claims that “routinely enable de novo structure solution of covalently attached RNA guest to beyond 3 Å overall resolution.” This statement does not reflect their results. In most cases the guest structures were not close to 3 Å and higher resolution was only achieved in limited regions. Overall, this is an interesting approach and one that could be very useful, but there are limitations that need to be discussed and a more candid description of the results is needed. In addition, some more specific points are listed below.

We agree the original language was perhaps over-zealous and have revised the statement (remove “routinely”). The relevant portions now read:

Abstract: “We now address this methodological gap by engineering 2- and 4-fold symmetric scaffolds that routinely enable *de novo* structure solution of covalently attached RNA guests to beyond 3 Å overall resolution.”

Introduction: “We now address this methodological gap by engineering 2- and 4-fold symmetric scaffolds that enable *de novo* structure solution of covalently attached RNA guests to beyond 3 Å overall resolution.”

1. It would help to have a table that lists the RNA guest molecules, the overall resolution attained and the resolution for the guests, not just the overall resolution. The important piece of information that needs to be highlighted is the resolution of the guests and completeness of the models for the guest molecules, not the overall numbers. The information is buried in the Extended Data. Instead, it should be a more prominent single Table.

We have added new Table 1 to the main text to indicate these points. This Table includes particle number, nominal resolution, C1/C2/D2 symmetry refinements.

2. The Methods section is long and detailed. Some of the details are not needed and the section could be more concise and focus on the aspects relevant to the scaffolding approach.

We have received guidance from another reviewer to provide additional details. Their concern was being able to apply the tool and having additional details. We have endeavored to make the methods more concise.

3. In the figures, such as Figure 4, it is not always clear where the guest molecule is found in the sequence and the structure. It should be made clearer.

We have updated the figure.

4. In the Extended Data Figures, it is necessary to add resolution to the color gradients. Best and Worst are not quantitative enough and do not allow proper assessment and comparison. Resolution numbers should be added to allow readers to assess the quality of the structures.

This “best to worse” coloring is recommended by recent cryoEM guidelines cited in the MS (<https://pubmed.ncbi.nlm.nih.gov/38358351/>, recommendation #14), as the numbers are not quantitatively reliable.

5. In several of the FSC curves shown there is a dip in the FSC. This is probably due to selecting a mask that is too tight or small. The authors need to correct or discuss this. It may reflect problems with data processing.

We have followed common masking guidelines in making masks for our refinements, which are described in the Methods (“CryoEM data analysis” section). In addition to this, during initial

data analysis, mask soft padding was varied to evaluate map quality to avoid this. Example masks overlaid with maps are now provided in Extended Figures 7 and 9. Dips in the corrected FSC curves (purple lines) are due to phase randomization.

6. As mentioned above, the Extended Figures should label the region corresponding to the guest molecules. It is not always obvious where the guests are as the orientations shown are not consistent.

We have fixed this in the relevant figures.

7. Extended Figure 2 mixes information on the Mango structure and the tRNA structure. All the Mango information belongs to Extended Figure 3 to separate clearly the two cases.

We have fixed this (now panel a in Fig. 3).

Reviewer #2 (Remarks on figshare data availability):

The data shows SEC traces, which are also shown in the paper. They appear to agree.

Reviewer #3 (Remarks to the Author):

This article by Jones and Ferre d'Amaré proposes a novel multimeric scaffolded method that is based on the structure of the RSV frameshifting element to routinely enable the de novo structure solution of covalently appended small RNAs by cryo-EM at resolution of 3 Å. This approach is particularly attractive as the scaffold used in this study is quite small in size but because of its ability of forming either dimers or tetramers, it still allows good atomic resolutions of various guest RNAs fused to it. The method proposed is quite novel and has the potential to contribute to solve the structures of many other small guest RNAs. This work definitely increases the tool box of RNA scaffolds that could be used for solving the structures of small RNAs that lack density for being able to be studied by cryo-EM. The experimental work and data analysis have been performed with much care and overall, the article is well written with many helpful figures. Because of its novelty, this article would certainly be suitable for publication in Nature methods. Nevertheless, it also raises few outstanding questions mentioned below that would be worth exploring further in order to increase further the potential of the proposed approach to be used by other cryo-EM laboratories.

Major comments

The guest RNA is inserted into a region (P3) of the RSV scaffold that is somewhat oriented away from the RSV core. However, the nature of the guest RNA seems to contribute somehow to the supramolecular assembly into either C2 or D2 symmetric particles. This is quite intriguing and it would be helpful if some rational could be proposed, especially from a methodological, rational engineering point of view. Is there any possible control over the formation of D2 versus C2 symmetric particles? What are the differences in conditions, processes that might favor the formation of C2 symmetric particles versus D2 symmetric particles or vice versa?

We have not yet been able to control C2/D2 symmetry. We have performed some studies addressing these questions (e.g., varying salt concentrations), but they demand a substantial amount of microscope time to thoroughly investigate multiple RNAs to have at least 3 guests. Our observation thus far is that higher RNA concentration and storage at 4°C both promote tetramer formation, but this is not the case for all RNAs. Likewise, the same RNA sample prepared for grids at similar concentrations formed tetramers in some cases and not others. These differences were observed between aliquots of the same sample as well as individual grids of the same sample blotted on the same day. Regardless of this heterogeneity, both C2- and D2-symmetric specimens are suitable for high-resolution structure determination, as we demonstrate.

We have also attempted to use the model of the D2-symmetric core to further engineer and stabilize the tetramer via mutagenesis (e.g., deleting flipped out C2575, mutating dimer/tetramer interface). These experiments have thus far been unsuccessful and are an area that merits further work. These experiments are complicated by the difficulty in observing tetramers consistently.

Accordingly we have now included a sentence (p10, first paragraph) stating

"At present, we do not have a full understanding of what experimental variables control C2- vs. D2-symmetric oligomerization."

The quinine aptamer and oxoG riboswitch are structurally related. Why does the 2OP quinine construct form mostly dimers while the 2OP-8oxoG assembles in particles with tetrameric D2 symmetry? Is this due to the 2bps difference in length of the connector stem P3 that connects the guest RNA to the scaffold?

We do not know. For the second question, this is an interesting hypothesis. However, for some grids we observe only dimers for 2OP-8oxoG. The sizes of 2OP-Quinine and 2OP-8oxoG are similar (monomers of 166 vs 160 nts). For both, the P3 helix does not participate in dimer/tetramer formation. Please also see response to query immediately preceding.

While the atomic resolutions of the quinine aptamer and oxoG riboswitch are outstanding, the resolution of the i-mango is not too great and it is somewhat surprising that the UA1 RNA protein complex is not able to be resolved with the proposed method. As the atomic structure of the i-mango aptamer and the UA1 RNP complex have been previously solved, what could be the factors behind the suboptimal resolution of these structures versus the two last ones (quinine aptamer and oxoG riboswitch)? Could this be due to variations in the way the folding protocols were performed, methodology, salt concentrations, presence of different ligands, or RNA engineering with P3 connectors being not optimal in length? While the answers to these questions might still be unclear as noted by the authors at the end of the discussion, it would be good to provide some clues in order to increase the likelihood of success of the proposed methodology once used by other laboratories.

The U1A result was puzzling to us as well, as we have crystallized RNAs with the same U1A sample. This RNA/protein sample was blotted at a relatively lower concentration of RNA compared to RNA-only samples, which may suggest U1A preferentially sticks to carbon. Other

datasets of the same complex (different sample not shown here) also failed to observe U1A bound. It would be a good future experiment to more systematically evaluate small RNA-binding proteins to determine if U1A is an outlier or this is a systematic problem that must be overcome.

For some RNAs, the resolution in solution will be attenuated due to the inherent flexibility of RNA. For these, one would expect maps to be poorer in flexible regions. The value of using a scaffold is that the scaffold is still high resolution in these cases, suggesting that the attached RNAs are lower due to their flexibility as opposed to experimental limitations (e.g., ice thickness, microscope, detector, RNA quality).

We have now included Extended Data Figures 7 and 9 to show data processing, including masks, which may help future use cases. Salt and RNA concentrations were similar throughout these experiments, aside for the 2OP-U1A sample. Future work will be needed to determine which guests have the best resolutions, based on parameters such as varying the length of P3.

Minor comments

Lane 116 p.7: it is 2OP rather than OP2.

We have corrected this.

Reviewer #3 (Remarks on figshare data availability):

The work has been done with great expertise and is clearly supporting the claims of the article.

This email has been sent through the Springer Nature Tracking System NY-610A-NPG&MTS

Response to Reviewers #2

Our responses are in red, the reviewers' original comments are in black, and the new changes are in blue.

Reviewer #1 (Remarks to the Author):

Thank you for the opportunity to review the revised version of this manuscript. In my previous review, I included a summary of the results and therefore will not articulate them here. My primary concerns in my review of the first submission were that the value of the symmetry was not clear or adequately discussed, some additional detail regarding the cryoEM workflow, the need for the fiduciary, and direct comparison of maps with and without the scaffold attached.

Overall, my concerns have been mostly adequately addressed and this is suitable for publication. I remain a bit skeptical that the symmetry is really adding much - I think it is almost certainly that the particles are larger and easier to align, but that is not a concern that precludes publication. I also think long-term application and benefit of the method will require much more optimization and examination of different RNAs beyond riboswitches, but this will likely be something that the field will work out in the coming years.

We agree that this is a question to be investigated moving forward. It may be more valuable if sample rigidity can be improved, allowing for refinement of the entire chimera (applying symmetry) without resorting to local refinement (not applying symmetry).

The one thing I remain a bit unhappy about is the lack of a map color coded by resolution. While I understand that a recent publication recommended using "highest to lowest" maps, such maps still give a range for the resolution using current statistical methods and are useful if even from a quantitative visualization and comparison. It is hard to justify referring in the text to the resolution of the final maps that the scaffold method obtained, and claim improvements of 0.1 Angstroms, and then not include a map because the numbers are not "quantitatively reliable." However, as this was a minor concern in my first review, I won't hold up publication over this.

We agree that 0.1 Å improvements are small, but the effect is consistent across the datasets, and the differences are larger for other map comparisons. We have now included the range of the local resolution maps in the captions for each one.

Reviewer #2 (Remarks to the Author):

The revised manuscript has addressed many concerns appropriately, but I still have some remaining concerns and suggestions. In general, the manuscript still exaggerates the resolution attained by the method. This is a serious concern as a read of the Abstract gives the false impression that it routinely attains resolutions over 3Å, which is not the case. See the comments below.

1. Table 1 Summarizes the results, including the resolution. The table is incomplete as it does not include the results for tRNA or U1A, both cases where the resolution was poorer. In addition, the

information that is most relevant, the resolution of the guest, is not included. The table should include the overall estimated resolution for host plus guest and in a separate column the resolution for the guest.

We assume that the reviewer means tRNA and Mango here, which had guests with poorer resolution. For these, the guest by itself was not refined as better alignment was obtained when a portion of the scaffold core was included in the reconstruction. Likewise, the cores were never refined alone. This improvement is likely due to the small size of Mango and poorer reconstruction of tRNA^{Asp}. Consequently, symmetry comparisons are only possible early on in analysis, so they were excluded.

Reporting the resolutions of “host with the guest” are of course going to be heavily weighted by the host, which is higher resolution. These will be the final refinements before symmetry expansion. The guest resolutions for these maps can be qualitatively estimated from local resolution maps.

2. Of the structures that are the test cases, U1A, tRNA, Mango, Quinine riboswitch, and 8oxoG riboswitch, only two of them achieve resolutions better than 3.0 Å for the guest (2.9 Å for both 2OP-Quinine and 2OP-8oxoG). The other 3 structures range from 3.7 – 3.5 Å and unknown (U1A). For this reason, the statement in the Abstract “... enable *de novo* structure solution of covalently attached guest to beyond 3Å overall resolution” is misleading. The sentence must reflect the results. In some cases the overall resolution of the guest extended just beyond 3 Å, but in most cases a more modest resolution was attained.

To address this, we added “for the best resolved guests” since this is, as we understand, the focal point of the concern. However, it is common to report the resolution like this for the best datasets. The new sentence in the abstract reads:

“We now address this methodological gap by engineering 2- and 4-fold symmetric scaffolds that enable *de novo* structure solution of covalently attached RNA guests to beyond 3 Å overall resolution for the best resolved guests.”

3. The most important thing that researchers want to know when assessing a method for solving a new RNA structure is whether it is possible to trace the structure *de novo*. It seems that this was only true in very few cases. This information should also be included in Table 1. This is important as the manuscript includes statements like “obtaining reconstructions at ~3.0 Å resolution for both ...” (Mango structures) followed by “Neither map was of sufficient quality to fully trace the intricate aptamer...”. Clearly the aptamer resolution was much worse than 3.0 Å, but an inexperienced reader may wrongly infer that the guest structure was at 3.0 Å and yet still untraceable.

We agree that this should be clear because this point hits the nail on the head. We have revised that section to now read:

As the structure of the unliganded RNA is unknown, we collected data on *apo*- and ligand-bound 2OP-Mango (Extended Data Fig. 3b,c), obtaining reconstructions at ~3.0 Å

overall resolutions for both although the guest portions are poorer (Fig. 3a,b and Extended Data Fig. 3d,e). Compared to 2OP-Mango bound to TO1-biotin (Fig. 3b), the guest portion of *apo*-2OP-Mango is poorly resolved, allowing only for a portion of the connecting helix to be modeled (Fig. 3a and Extended Data Fig. 3f,g). Neither map was of sufficient quality to fully trace the intricate aptamer *de novo* (Fig. 3c), but the ligand-bound cryoEM map is consistent with the overall crystal structure of *iMango*-III A10U, which can readily be rigid-body docked (see Methods).

4. The reason why there is a recommendation not to use numerical values for local resolution estimate maps is that the methods to calculate local resolution are unreliable, not because this is unwanted or irrelevant information (“The community consensus appears to be that estimated local-resolution values are not quantitatively reliable. Thus, the global resolution estimate could be supplemented by a coloured local-resolution map accompanied by a colour legend labelled not with specific Ångström values, but rather ‘better’ (blue) to ‘worse’ (red) resolution. Finally, it was recognized that, for cases where an atomic model is available, a visual depiction of local resolution across the amino-acid or nucleotide sequence would be a valuable addition to future validation reports, once robust algorithms that produce such a mapping become available and are widely accepted in the community.”) I suggest that at the very least the color legend should show the resolution value estimated for the structure. For example, in Extended Data Figure 3, the color ramps should have best and worse plus the actual estimated resolution values, in this case 2.9 Å and 3.0 Å marked in the color ramps. In this way the readers can have a reference for what color corresponds to the estimated resolution.

We have now reported the scales from blue to red for each local resolution map in the relevant figure captions (Extended Data Figures 1, 2, 3, 4, 6, 8).

5. I still think that in every figure the guest molecule should be clearly marked. Referring to Extended Data Figure 3 again, the Mango portion of the structures should be boxed to make it very clear where this portion is. This is very important to assess the resolution of the guest portion of the molecule in relation to the host.

We have added dashed boxes to the relevant figures with guests (Extended Data Figure 3, 4, 6, 8 and Figure 2) and indicated this in the relevant figure captions.

6. There are some minor errors that could be removed by careful editing. For example, in the caption of Extended Data Figure 5, it says “a map threshold of 10.” I think they mean a map threshold of 10 Sigma.

This is a good catch. We have revised each of these in the Extended Data Figure 5 caption:

e, Views of locally refined core map from tilted data, contoured at 8 σ in Pymol. **f**, Views of locally refined aptamer map from tilted data, contoured at 10 σ in Pymol.

Reviewer #3 (Remarks to the Author):

All the comments of the referee have been addressed.

Reviewer #3 (Remarks on figshare data availability):

None

This email has been sent through the Springer Nature Tracking System NY-610A-NPG&MTS